# Roles of Sulfites in Reverse Osmosis (RO) Plants and Adverse Effects in RO Operation

**DOI:** 10.3390/membranes12020170

**Published:** 2022-01-31

**Authors:** Yasushi Maeda

**Affiliations:** LG Chem Japan Co., Ltd., Kyobashi Trust Tower 12F, 2-1-3 Kyobashi Chuo-ku, Tokyo 104-0031, Japan; jpmaeda@lgchem.com; Tel.: +81-3-5299-4530

**Keywords:** bisulfite, metabisulfite, reverse osmosis, dechlorination, ORP, chloramine, chlorine dioxide, preservative, storage, shock treatment, oxidation, degradation, auto-oxidation, heavy metals, radical, a chelating agent, biofouling, trigger, biocides, AOC, cleaning

## Abstract

More than 60 years have passed since UCLA first announced the development of an innovative asymmetric cellulose acetate reverse osmosis (RO) membrane in 1960. This innovation opened a gate to use RO for commercial use. RO is now ubiquitous in water treatment and has been used for various applications, including seawater desalination, municipal water treatment, wastewater reuse, ultra-pure water (UPW) production, and industrial process waters, etc. RO is a highly integrated system consisting of a series of unit processes: (1) intake system, (2) pretreatment, (3) RO system, (4) post-treatment, and (5) effluent treatment and discharge system. In each step, a variety of chemicals are used. Among those, sulfites (sodium bisulfite and sodium metabisulfite) have played significant roles in RO, such as dechlorination, preservatives, shock treatment, and sanitization, etc. Sulfites especially became necessary as dechlorinating agents because polyamide hollow-fiber and aromatic thin-film composite RO membranes developed in the late 1960s and 1970s were less tolerable with residual chlorine. In this review, key applications of sulfites are explained in detail. Furthermore, as it is reported that sulfites have some adverse effects on RO membranes and processes, such phenomena will be clarified. In particular, the following two are significant concerns using sulfites: RO membrane oxidation catalyzed by heavy metals and a trigger of biofouling. This review sheds light on the mechanism of membrane oxidation and triggering biofouling by sulfites. Some countermeasures are also introduced to alleviate such problems.

## 1. Introduction

Reverse osmosis (RO) is a liquid-phase pressure-driven separation process in which applied transmembrane pressure causes selective movement of solvent against its osmotic pressure difference [1]. RO is now ubiquitous in water treatment and has been used for various applications, including seawater desalination, municipal water treatment, wastewater reuse, ultra-pure water (UPW) production, and industrial process waters, etc. Furthermore, RO is anticipated to contribute to the United Nation’s Sustainable Development Goals (SDGs), especially in Goal 6: Clean Water and Sanitation. Then, in Goal 6.a, the following actions are raised: expand international cooperation and capacity building support to developing countries in water and sanitation-related activities and programs. These include water harvesting, desalination, water efficiency, wastewater treatment, recycling, and reuse technologies [2,3].

Regarding RO membrane development, more than 60 years have passed since UCLA first announced the development of an innovative asymmetric cellulose acetate RO membrane in 1960. Furthermore, new generation polyamide hollow fiber RO and thin-film composite (TFC) aromatic polyamide RO membranes were developed one after another in the early 1970s and 1977. As a result of continuous improvements, the TFC RO membrane performance has been greatly improved, and it is now widely used for a variety of applications. As for the future membrane desalination technology, three technologies were raised in National Geographic, April 2010 [4]. These three technologies promised to reduce the energy requirement of desalination up to 30% are forward osmosis, carbon nanotubes, and biomimetics. Among those, nanoporous membranes, including porous graphene, carbon nanotubes, and graphene oxide, etc., attracted much attention from academic researchers [5]. However, it does not seem easy to produce commercial-based defect-free RO membranes with nanoporous materials. A way of overcoming material limitations for RO applications is to utilize composite materials comprising nanoporous materials within a polymer matrix. The use of thin-film nanocomposite (TFN) membranes for water purification was first described for BWRO membranes by Jeong et al. [6]. After that, many research works on the TFN membranes have been conducted [7,8].

An RO system typically consists of five major unit processes: (1) intake system, (2) pretreatment, (3) RO system, (4) post-treatment, and (5) effluent treatment and discharge system, as illustrated in Figure 1. RO membranes and elements are critically important to separate water from organic and inorganic impurities in the RO system. Several RO membranes exist, such as aromatic polyamide, and cellulose triacetate, etc., and are fabricated into the spiral-wound, hollow fiber, tubular, plate and frame elements. Among them, thin-film composite (TFC) and thin-film nanocomposite (TFN) spiral-wound elements have been commonly used in water treatment. However, a series of pretreatment is necessary to supply feedwaters to RO elements and meet specific requirements for the spiral wound elements, such as silt density index (SDI) < 5 and residual chlorine < 0.1 mg/L, etc.

It is observed that a variety of chemicals have to be used in each process steps shown in Figure 1. In the intake system, chlorine is sometimes applied continuously or intermittently to protect the intake and pretreatment equipment from bacteria and algae growth. The following chemicals are used in the pretreatment step: coagulants, flocculants, dechlorination agents, and antiscalants, etc. When using low-pressure membranes (MF/UF) as the pretreatment, backwash chemicals, such as sodium hypochlorite (NaOCl) and acid/caustic for chemical-enhanced backwash (CEB), are used. In the RO system, cleaning in place (CIP) chemicals, RO element storage chemicals (preservatives), and biocides are used.

However, it should be noted that the use of these chemicals depends strongly on the feedwater characteristics and operating conditions. For example, some well water treatment plants are only equipped with cartridge filters with minimal chemical dosage [9,10,11]. It is reported that the 360 m^3^/d capacity BWRO plant in Las Palmas, Canary Islands, Spain, has been operating for more than nine years by only dosing 6 mg/L of antiscalant [12]. Similarly, Lagartos et al. [13] reported Malta’s Pembroke seawater desalination plant. The plant can produce 54,000 m^3^/d of water. The water intake comes from beach wells with a silt density index below 1. The pH is adjusted with sulfuric acid down to 6.7 to protect the pipework and prevent scaling. After pH adjustment, cartridge filters are installed upstream of the RO unit. It is reported that the cleaning in place (CIP) frequency varies between 6 and 10 months, depending on the train condition and time of operation.

These chemicals might be categorized into the following six (6) items [14]: (1) cleaning agents, (2) dechlorinants, (3) biocides, (4) pH adjustors, (5) coagulants/flocculants, and (6) antiscalants. As for dechlorinants, either sodium metabisulfite (SMBS) or sodium bisulfite (SBS) is used, and it was estimated as roughly 12–15% of the membrane chemicals market. However, SBS is used for many other applications in the RO unit processes, as shown in Figure 2.

The most critical role of SBS is dechlorination. As the TFC/TFN membranes are less tolerable to chlorine, the residual chlorine must be removed prior to entering the RO unit. The next key application is a use for an RO element preservative for shipping elements and during plant shutdown. Several other SBS applications include deoxygenation, shock treatment as a biostatic agent, and CIP chemical, etc. Thus, SBS can be considered an essential chemical for RO processes. However, some adverse effects have been reported. For example, it was reported that under specific conditions, i.e., heavy metals, dissolved oxygen, etc. SBS degrades RO membranes [15], or SBS triggers biofouling when overdosing [16].

In terms of dechlorination and preservative roles, there have been many reports on how to use SBS and control its dosing amount. However, fewer reports were observed for the membrane degradation and inducing biofouling. Therefore, this review article aims to shed light on some adverse effects of SBS and identify their mechanisms in addition to common application fields.

## 2. Chemical Properties and Handling Precautions of Sodium Bisulfite (SBS)

SBS is a chemical compound with the chemical formula NaHSO_3_ that has a reduction ability. Thus, it is used to remove residual chlorine in water/wastewater and industrial applications. It is also used as an oxygen scavenger in boiler water treatment. In the food industry, SBS is used as a preservative. Some fundamental chemical properties are shown in Table 1.

SBS is a weakly acidic species with a pKa of 6.97. Thus, SBS exists as a mixture with sodium sulfite in the neutral pH range as shown below:HSO_3_^−^ ↔ SO_3_^2−^ + H^+^(1)
SBS is available as a solution of various concentrations or is produced by dissolving SMBS (Na_2_S_2_O_5_). When SMBS is dissolved in water, SBS is formed:Na_2_S_2_O_5_ + H_2_O → 2NaHSO_3_(2)
SMBS solution has a pH of 4.6 at 1.0% (by weight) solution strength [17]. It is demonstrated that sulfur dioxide (SO_2_) vapor pressure is increased at lower pH of less than 5.5 for a 30% active SBS solution [18]. It is also demonstrated that SO_2_ generation begins at pH 7.0, and a fair amount of SO_2_ gas is generated below pH 4.0 according to the following equilibrium [19].
SO_2_ + H_2_O ↔ HSO_3_^−^ + H^+^(3)

If SMBS is used to produce SBS, SO_2_ is generated when mixing with water. Therefore, a dilution tank must have a vent [20]. If more than one (1) dilution tank is installed, they will be interconnected and extracted to a safe location. Furthermore, SBS reacts with oxygen during storage. The deoxygenation reaction increases sulfate concentration and decreases pH, further inducing SO_2_ off-gas generation from the storage tank. Therefore, an air extraction system must be installed in the area and a vent must be directed outdoors.
2HSO_3_^−^ + O_2_ → 2SO_4_^2−^ + 2H^+^(4)

Releasing hazardous fumes can be reduced by using sodium sulfite (Na_2_SO_3_) solution or increasing the pH of the SBS solution. However, it is limited in its day tank solubility to about 12%. Even at lower concentrations, constant mixing within the day tank will be required [21].

The food-grade SMBS powder has a shelf life of approximately 6–12 months. However, as the SMBS/SBS solutions are not stable to air and react with oxygen, the shelf life of the solutions is shortened depending on concentrations. Therefore, the following guideline is suggested from RO membrane manufacturers as shown in Table 2.

The solubility of SBS is significantly reduced at a temperature of less than 10 °C [18]. Thus, higher concentrations tend to crystallize at relatively warm temperatures (≤6 °C), causing blockages in dosing pump suction and delivery lines, and lower concentrations of around 20 wt% are sometimes preferred for this reason [25]. Thus, it is recommended that SBS solution should always be stored in a temperature range of 15–35 °C [18]. Kunisada et al. [26] encountered the SBS crystallization problem for an 800 m^3^/d pilot plant operation in Chigasaki, Japan. When the temperature was close to 0 °C, SBS was crystallized, which caused blockage of the chemical injection line and pump. Therefore, several measures were taken but could not resolve the issue completely. Finally, it was decided that the 35% SBS solution was diluted to 30% by installing an additional 4 m^3^ tank.

In case of spillages and safely disposing of SBS solutions, special care has to be taken. It is guided that any spillages should be neutralized with soda ash to prevent SO_2_ emission and then be oxidized to neutral sulfate with sodium hypochlorite [25]. On some occasions, higher concentration SBS solutions are discharged from RO systems, including startup time after RO unit preservation and shock treatment during RO operation. In some RO plants, aeration is applied to neutralize SBS [27,28,29,30]. For example, in an RO plant producing boiler make-up water, a wastewater treatment unit was installed. When the RO plant is shut down, the RO train is preserved with about 500 mg/L of SBS. Therefore, when the RO starts operation, a large amount of SBS is discharged into the brine. An aeration unit was equipped to address an issue of a regulated COD. During aeration, the pH is adjusted to 6.0–8.5 with caustic soda. As the aeration is proceeding, SBS concentration reduction is stopped at about 10 mg/L. However, since the COD is reached about 1.5 mg/L at this stage, it can be released. A similar treatment was implemented in the 40,000 m^3^/d seawater desalination plant in Okinawa, Japan [28].

## 3. Removal of Oxidative Disinfectants: Chlorine, Chloramine, Chlorine Dioxide, and DBNP

Chlorine disinfection has been applied to protect intake facility and pretreatment equipment from biological growth and reduce the risk of biofouling in RO modules/elements. When RO membranes were first commercially used, cellulose acetate (CA) was a primary RO material. As the CA membranes have a certain degree of chlorine resistance up to a maximum of 1.0 ppm [31], much attention was not paid to dechlorination. However, soon after the aramid hollow fiber RO membrane was put into practical use, the dechlorination process became an important issue because of its poor chlorine resistance [32]. Similar dechlorination conditions were applied to the newly developed TFC membranes, although the TFC polyamide membranes were considered to have some chlorine tolerance (1000–2000 ppm·hr) [23]. Even though dechlorination is of critical importance for RO operation and maintenance (O & M), it is reported that nearly 18% of RO elements failure was attributed to membrane oxidation during element autopsy studies [33,34]. Thus, dechlorination is now a crucial pretreatment step due to their insufficient chlorine resistance.

Currently, low-pressure (LP) membranes have been used as pretreatment of RO. In this case, a high chlorine concentration is used for backwash water and CIP chemical when the LP membranes are used. Thus, residual chlorine in the LP membrane permeate has to be removed before entering the RO. Furthermore, some other types of disinfectants, such as chloramine and chlorine dioxide, have been considered to address disinfection by-product (DBPs) formation by chlorine. Generally, these disinfectants have less oxidative power and might be dosed to RO continuously or intermittently. However, it is reported that the disinfectants oxidize RO membranes under specific conditions. For example, it is known that chloramine is converted to bromamine in seawater and that bromamine causes RO membrane oxidation [35]. Thus, when chloramine and chlorine dioxide are used in the RO process, those oxidants may need to be removed.

As mentioned, several types of oxidants are used in pretreatment, CIP, and disinfection, etc. In addition to preventing RO membrane oxidation, the residual oxidants must be removed prior to discharge to protect the watershed environment. In this section, oxidative chemical removal technologies with sulfites will be discussed.

### 3.1. Dechlorination

Dechlorination has been achieved by an activated carbon (AC) bed or sulfite chemicals in RO processes. However, carbon filtration is typically not recommended for dechlorination of RO feed water unless the concentrations of organics are high enough to warrant its use and other circumstances prohibit the use of sulfites [24,36]. Usage of AC filters has the following concerns in RO O&M: aiding the growth of microbes and sloughing off carbon fines. Thus, sulfite compounds, sodium sulfite (Na_2_SO_3_), SBS (NaHSO_3_), and SMBS (Na_2_S_2_O_5_) are used for dechlorination. These sulfites react with chlorine as follows:Sodium sulfite: Na_2_SO_3_ + Cl_2_ + H_2_O → Na_2_SO_4_ + 2HCl(5)
Sodium bisulfite (SBS): NaHSO_3_ + Cl_2_ + H_2_O → NaHSO_4_ + 2HCl(6)
Sodium metabisulfite (SMBS): Na_2_S_2_O_5_ + 2Cl_2_ + 3H_2_O → 2NaHSO_4_ + 4HCl(7)

Based on these reactions, theoretical dosages for different sulfites are summarized in Table 3.

SBS and SMBS have been commonly utilized in RO dechlorination among those sulfites. When using SMBS, non-cobalt catalyzed and food-grade quality SMBS should be used. It is reported that cobalt-catalyzed SBS for dechlorination resulted in the degradation of polyamide membranes [31]. Regarding the necessary dosing amount in the field, a stoichiometric dosage of SBS was insufficient for complete dechlorination. Thus, an excess of the stoichiometric dosage (mg dechlorination agent/mg Cl_2_) is needed. However, it was unclear how much extra dosing should be applied to actual plants. A few research works were conducted by focusing on this issue and measuring the reaction kinetics of various reducing agents under different stoichiometric dosing rates [38,39,40]. The findings suggest that the three stoichiometric dosage of SBS was successful in achieving complete dechlorination. It was also found that organic and inorganic matter may be responsible for inhibiting dechlorination at low stoichiometric dosages of SBS [39]. Apart from these experimental results, 10% excess dechlorinating chemicals are suggested for common water and wastewater treatment [37]. However, in the case of RO application, a higher amount of sulfites dosage has been recommended due to a concern about membrane oxidation and stability of sulfites during storage. Almost all RO-related books and manufacturers’ technical bulletins state an appropriate amount of SBS dosing for dechlorination. The suggested dosing amount so far is summarized in Table 4.

Typically, 2–3 mg of SMBS are suggested to remove 1 mg of chlorine. Using three to five times the stoichiometric amount of SBS was suggested for the aramid hollow fiber RO [43]. In Table 4, the stoichiometric amount ratio (dosing amount/stoichiometric amount) is also listed. When determining the actual dosage, the stoichiometric amount ratio might be considered a safety factor [36]. The following safety factors are generally applied: 1.2–2.0 [46] and 1.5–2.0 [36]. One technical bulletin mentioned that more SMBS might be required for seawater when dissolved oxygen is present [23]. This suggestion may have referred to an uncommon event in the Middle East. The Umm Lujj 2 Desalination Plant is located on the Red Sea Coast 154 km north of Yanbu. The Umm Lujj 2 was designed to produce 4400 m^3^ per day of drinking water and started in 1986. A 28-day trial using 0.5 ppm chlorination and dechlorination accompanying 5 ppm SBS was unsuccessful [47]. Even with adopting a higher safety factor of 6.8, damage to chlorine-sensitive membranes was found, and this degradation resulted in premature failure of membrane modules [48]. A membrane autopsy revealed that the membrane was attacked by halogen compounds [49]. Osta et al. [47] attributed this phenomenon to a fast reaction with oxygen as one of the reasons due to: (a) metals in seawater serving as catalysts, (b) high ionic strength, (c) specific pH, (d) high bicarbonate concentration, and (e) high temperature. As for heavy metals, the raw seawater of the Red Sea was analyzed along with troubleshooting efforts for CTA membrane oxidation in the Jeddah Phase I plant [50]. A higher copper ion concentration of 1.8 ppb was detected than the standard concentration of less than 0.2 ppb. Therefore, special attention should be paid when expecting a higher level of heavy metals from raw seawater and coagulants (impurities). In this case, residual chlorine may attack RO membranes rapidly.

Thus far, the safety-dosing amount of SBS/SMBS for preventing RO membrane oxidation was discussed. However, as mentioned in Section 8, overdoing sulfites may have adverse effects, e.g., membrane oxidation, biofouling, etc. [51]. Thus, the residual SBS concentration should be carefully controlled. In addition, it is said that the over-injection of sulfite causes an increased breakdown of dissolved oxygen in the water. This kind of environmental stress increases the potential for a heavy growth of slime-forming species of bacteria, which can quickly foul an RO system. Byrne [52] pointed out that this potential can be minimized by maintaining a residual sulfite concentration greater than zero but less than 2 mg/L as SBS.

### 3.2. Dechlorination Point Considerations

The rate of dechlorination is rapid in laboratory experiments. At three times the stoichiometric amount of SBS/SMBS, dechlorination time reaching 0.02 mg/L of residual chlorine is 37 s for 1 mg/L of initial chlorine concentration [38]. Other reference articles reported a similar completion time of 15–20 s [20,37]. Thus, it is expected that an excess of the stoichiometric dosage of sulfites could achieve complete dechlorination within less than 1 min [53]. The dechlorination reaction requires mixing to ensure completion. Therefore, proper in-line mixing is needed, which preferably includes a static mixer [20]. When SBS is dosed after cartridge filters (CFs), the SMBS solution should be filtered through a separate cartridge before being injected into the RO feed [41].

The next issue is where SBS should be injected either before or after a CF. Until the 1980s, it was recommended that SMBS is injected prior to the CFs [22,23]. However, in the 1990s, membrane manufacturers began suggesting that the SBS injection point is set after the CF [43,54]. One RO manufacturer recognized that the optimum injection point of SBS is at the suction of a high-pressure pump and started recommending the new SBS injection point location for existing RO plants as well as for new projects [43]. It was said that this dechlorination injection point location would minimize or eliminate biological fouling in all piping before the high-pressure pump suction. However, to implement this technology, a potential difficulty assuring no chlorine entrance to the RO modules was addressed, as redox meters have a response time of 45 to 60 s. The hold-up time for the feed between the SBS addition and the RO modules is less than this value. Thus, there is a risk that some residual chlorine could enter the RO modules before the alarm is given. However, few system failures were reported for the aramid hollow fiber RO. Such practice was implemented in some plants, such as the Dhekelia SWRO plant in Cyprus [55].

Shifting the SBS injection point after the CFs certainly positively affects suppressing differential pressure increase of the CFs and reducing filter exchange frequency [56]. However, in terms of suppressing the RO membrane biofouling, its effectiveness is not apparent. Saeed [57,58] reported contradictory results. In the test conducted at the Ar-Birk SWRO plant, bacterial generation (doubling) time was used to evaluate biofouling potential. The generation time was higher (lower multiplication capacity) when the SBS dosing point was before the CF. On the other hand, the generation time was decreased significantly, reflecting higher multiplication capacity and higher biofouling potential when the SBS dosing point was moved to after the CF. This observation means that the closer the SBS dosing point location to the RO membranes, the greater the biofouling potential and biofilm formation. This correlated well with operational data of doubling membrane-cleaning frequency when the SBS dosing point shifted to after the CF [57]. However, it should be noted that the chlorine concentration used to disinfect the feed to the Al-Birk plant was 4 ppm at the intake and 1–1.2 ppm after the filters. The residual chlorine was removed by dosing an average of 6 ppm of SBS, much higher than the stoichiometric amount [59]. Thus, the effect of excessively added SBS may have to be considered when interpreting the results.

As observed in the Al-Birk test, it seems challenging to solve the problem alone by changing the injection point. Such cases have been reported in several plants. For example, the Gabès 22,500 m^3^/d BWRO plant in Tunisia was installed in June 1995. The pretreated water was initially dechlorinated using SBS before the cartridge filter preceding each high-pressure pump. Soon after the plant startup, severe biological fouling occurred in the RO units. Change of the point of injection of SBS upstream to the downstream of the CF allowed eliminating the problem only in the filter and stabilizing the pressure drop through the filter. Such a biofouling problem has continued until the chlorination procedure was changed to intermittent chlorination/dechlorination method [60].

A similar phenomenon was also observed in the Arcadia Water Treatment Plant in Santa Monica, California. The BWRO plant treats local groundwater to provide up to 38,000 m^3^/d of treated water as part of the City of Santa Monica’s drinking water supply. The original plant configuration included dosing of SBS immediately upstream of the CFs to quench any residual chlorine from the upstream greensand filters. Following the identification of biological growth in the CFs, plant staff reconfigured the SBS dosing downstream of the CFs, allowing chlorine residual to disinfect the CFs effectively. However, while the biofouling was arrested at that location, it spread to the downstream RO membranes themselves [61]. Therefore, in this plant, chloramine addition was implemented to tackle the biofouling. It was reported that this new disinfection protocol resulted in a significant reduction in biofouling.

The following example is the seawater desalination plant with an 18,000 m^3^/d capacity in Santa Barbara, Curacao [62]. Chlorine is dosed in the beach clear well. SBS is injected after a CF when shock pre-chlorine is performed. After an initial lag period, the differential pressure (DP) increase becomes more rapid. During the first 15 months, the plant had to conduct CIP five times. The autopsy data demonstrates that this DP increase is due to biological growth on the membrane. To reduce the rate of biofilm growth, a program of weekly, overnight biocide soaks with a commercial non-oxidative biocide was implemented. The result was a significant reduction in the rate of DP increase. A subsequent attempt was to control SBS dosing. In the original design for the Santa Barbara plant, SBS dosing was based on the free chlorine level anticipated during the regular shock chlorination. This practice ensures that no chlorine reaches the membrane; however, it also results in an excessive SBS residual in the feed. Then, the SBS addition program was modified to reduce the SBS excess. After the cleaning was performed, the rate of DP increase was immediately and positively affected by the change in SBS addition.

As observed in the case studies above, it was found that shifting the SBS injection point alone does not ensure a biofouling-free operation, even though this practice has a positive effect on reducing the DP increase rate in the CF. Thus, other measures have to be considered to control biofouling. These include intermittent chlorination, chloramine/chlorine dioxide disinfection, and minimizing SBS dosing amount, etc.

### 3.3. Monitoring Dechlorination

Dechlorination has been monitored by either a chlorine analyzer or an oxidation-reduction potential (ORP) meter [20,22,41]. In addition, measuring residual SBS concentration is helpful to avoid overdosing. It seems that DuPont first considered applying the ORP for monitoring residual chlorine to RO systems. Their study indicated that ORP could be useful in an indication of the level of reduction of oxidant (chlorine) used for disinfection in seawater [63]. However, at that time, the ORP technology was not mature enough, and readings demonstrated extreme excursions. After that, along with technological improvements and actual plant data accumulation, it became common to equip with an ORP meter alone or together with a chlorine meter in seawater desalination. In a particular case, it was reported that two ORP meters and one chlorine meter were installed to ensure chlorine removal in the Shuqaiq desalination plant [64,65].

The ORP reading is rapidly increased by adding a small amount of chlorine. Figure 3 shows the ORP changes at low residual chlorine concentration [66]. It is observed that the ORP readings are increased by nearly 200 mV when the residual chlorine concentration increases to 0.02 mg/L as Cl_2_. This ORP characteristic is a practical background that a certain level of ORP reading is used a high (H) alarm signal or high–high (HH) alarm signal. When detecting an H-alert, it might be possible that the SBS pump doses a higher amount of SBS to address the increased chlorine in the feed water [67]. If the ORP value reaches HH level, the plant should be shut down until the oxidant concentration can be reduced to a safe value [20,22,54].

In terms of the H and HH alarm levels, several readings have been proposed by membrane suppliers and experts in this area. Table 5 summarizes the proposed H and HH threshold ORP value.

A slight difference in H and HH values can be observed. One manufacturer proposes the pH-dependent H and HH values, as the ORP reading depends on feed pH. Thus, one may need to consider adding or reducing 50 mV for every one (1) change in pH [66]. Although the listed H and HH values are not limited to specific water types, it might be natural to consider that they are mainly applicable to surface seawater, as the pretreated seawater conditions are not significantly varied. For example, in the Okinawa SWRO plant, the 250 mV of ORP was set as the HH alarm [28].

It is known that the ORP value depends on various factors, such as water sources (groundwater, surface water, TDS, ions, etc.), pH, dissolved oxygen, temperature, and the organics present in the water [46,72]. In addition, it is pointed out that the absolute reading of the ORP meter may fluctuate due to factors, such as electrode contamination, due to continuous use and fluctuations in the manufacturing factors of the ORP electrode itself [73]. These characteristics mean that the ORP value completing the chlorine removal varies with feed water types and physicochemical conditions, as schematically shown in Figure 4.

In one case, it was reported that once the ORP reading started to plateau between +350 and +400 mV, similarly to “Feed Water 1”, all of the chlorine reacted with SBS. Therefore, if one tries to follow the guideline H values, e.g., 300 mV shown in Table 5, dosing extra SBS beyond the ORP plateau would be excessive [72,74].

One of the most critical factors affecting the ORP reading is the feed pH. It is well known that ORP decreases with increasing pH. Furthermore, a pH-dependent equilibrium between hypochlorous acid (HOCl) and hypochlorite ion (OCl^−^) affects the ORP changes. At pH = 6.0, HOCl is dominant, and at pH = 9.0, OCl^−^ is a dominant species. These two have the standard electrode potentials of 1.48 V and 0.81 V, respectively.
HOCl + H^+^ + 2e^−^ → Cl^−^ + H_2_O(8)
ClO^−^ + H_2_O + 2e^−^ → Cl^−^ + 2OH^−^(9)

The reported dechlorinated water ORP at pH = 10.0, where the second-pass feed pH was increased to improve boron rejection, showed about 30–40 mV [15]. Byrne [75] sounded the following alarm related to the ORP fluctuations with pH. When ORP is used to control SBS dosage, the results may be disastrous if the RO permeate returns to an upstream feed tank when process water is not demanded. During times of minimal usage, the ratio of RO permeate in the blended feed is increased. Added SBS will have an increased impact on the water pH and cause it to drop. The declining pH will cause the ORP reading to increase even if no chlorine is present. The control system will respond by adding even more SBS, and the SBS injection pump will eventually max out on its dosage.

The next unclear point about the ORP measurements is the effect of salinity or total dissolved solids (TDS). There is a lack of knowledge on how salinity might influence ORP during chlorination/dechlorination. Xie et al. [76] titrated chlorinated water (using sodium hypochlorite, NaOCl) with SMBS. The ORP was monitored for waters of different salinities prepared by diluting seawater (TDS, 33,800 mg/L, pH 8.2, Singapore) with deionized water (TDS, 50 mg/L). The most critical parameter for RO dechlorination might be the endpoint ORP. From their results, the following two key findings can be drawn. First, before adding the titratant, the raw water had ORP values that varied from 270 mV for deionized water to 54 mV for seawater. Similarly, when injecting the same amount of NaOCl, the seawater sample demonstrated the lowest ORP value. The endpoint ORP difference between seawater and 25% seawater was nearly 150 mV. Second, the endpoint ORP value was increased by increasing the initial NaOCl dosing amount. It is reported that the endpoint ORP is increased from 75 mV to 250 mV by increasing the initial chlorine dosage from 1 to 5 mg/L NaOCl in seawater.

Based on these reported data, it is crucial to experimentally determine ORP set points (H and HH) in each plant for controlling or monitoring the residual chlorine. Tate [77] mentioned that ORP setpoints would vary from site to site, thus an experienced technician should run titration tests to determine the optimal setpoint. For instance, ORP is set 30 to 50 mV lower than that at which 0 ppm free chlorine is achieved. In addition, it is critical to measure the residual chlorine by a chlorine meter as needed. Lindgren and Casey [78] suggested calibrating the ORP sensors to measure free chlorine residual values, ensuring that the TFC membranes do not see free chlorine. A portable test kit is used once per week to measure the free chlorine residual to verify the ORP monitor is functioning properly. This kind of practice is crucial to avoid any abnormal ORP events and membrane oxidation. For example, in the Tampa Bay desalination plant, unusually high ORP values within the feed to the RO trains with no free chlorine concentration were detected, resulting in overdosing SBS (20 ppm) [79].

Up to this point, monitoring ORP and measuring free chlorine methods are discussed to ensure the chlorine-free feed supply to RO. However, as observed in Figure 4, the ORP reading is relatively insensitive to excessive SBS concentration. Thus, RO plants tend to overdose on SBS. It is indicated [70] that the excess amounts of SBS may lead to rapid membrane oxidation from catalytic reactions when the feed water contains transition metals (e.g., Co, Cu, Mn, etc.) or membranes are fouled with the transition metals. In addition, the excess amount of SBS may lead to biofouling from the growth of sulfate-reducing bacteria, severely deteriorating the membrane performance. Thus, the RO membrane supplier recommends keeping the residual SBS in the feed water below 1 mg/L [70]. Byrne [52] mentions that the biofouling potential can be minimized by maintaining a residual sulfite concentration greater than zero but less than 2 mg/L as SBS. Therefore, it is imperative to measure the residual SBS concentration to conform to these targets.

Two sulfite analysis methods are indicated in the standard methods (4500-SO_3_^2−^ SULFITE): the iodometric method and phenanthroline method [80]. The iodometric titration method is suitable for relatively clean waters with concentrations above 2 mg SO_3_^2−^/L. However, following the evolution of sulfite from the sample matrix as SO_2_, the phenanthroline colorimetric determination is preferred for low sulfite levels. In this method, an acidified sample is purged with nitrogen gas, and the liberated SO_2_ is trapped in an absorbing solution containing ferric ion and 1,10-phenanthroline. Ferric iron is reduced to the ferrous state by SO_2_, producing the orange tris(1,10-phenanthroline) iron (II) complex. After excess ferric iron is removed with ammonium bifluoride, the phenanthroline complex is measured colorimetrically at 510 nm. In addition, as the RO industry is familiar with chlorine analysis by colorimetry and ORP, the back-titration with chlorine might be an option.

### 3.4. Precautions for Integrated Membrane System (IMS)

Various types of hybrid membrane processes have been applied to water and wastewater treatment. The hybrid membrane process is the combination of a conventional unit operation, such as distillation, evaporation, or electrodialysis (ED), with a membrane process, such as RO [81,82,83]. The low-pressure (LP) membrane and NF/RO combination have played important roles in municipal water, wastewater treatment, and seawater desalination. In the late 1990s, AWWARF and USEPA funded the project “Integrated multi-objective membrane systems for control of microbial and DBP precursors” [84]. Originally, the concept, referred to as the integrated membrane systems (IMS), covered a wider process area: (advanced) pretreatment processes combined with NF or RO. However, the IMS was narrowed down later to a combination process of LP membrane and NF/RO [85].

LP membranes, including microfiltration (MF) and ultrafiltration (UF), have been widely used as pretreatment to RO. In the LP membrane process, chlorine, usually sodium hypochlorite (NaOCl) solution, is used for cleaning steps in addition to other chemicals. The following three cleaning methods using NaOCl are commonly utilized and summarized by Gilabert-Oriol et al. [86].

Backwash: Backwash conducts to clean the fibers and, consequently, reduce the transmembrane pressure (TMP) accumulated during filtration. NaOCl has been the most widely used, and its typical range is 3–20 mg/L with a median of 10 mg/L [87].Chemical-enhanced backwash (CEB): the CEB occurs once or twice per day, is characterized by taking longer than the backwash, and is conducted by the use of chemicals. For example, NaOCl concentration is at 20–500 mg/L with a median of 150 mg/L.Cleaning in place (CIP): CIP occurs once every couple of months and is characterized by its longer duration (a few hours typically) and higher chemical concentrations used compared with CEB. NaOCl is used at elevated concentrations (up to 4000 mg/L with PVDF fibers) for oxidative cleaning.

Thus, it is crucial that residual chlorine does not reach the RO system when using the LP membranes as pretreatment. Busch et al. [88] mentioned that a few erroneous exposures could totally exhaust the limited chlorine tolerance of SWRO membranes. Such cleaning practices add a very critical and risky variable to the IMS.

Many pilot tests were conducted in the 2010s to identify the benefits of using the LP membranes compared with conventional pretreatment. There were several reports that RO membrane oxidation occurred due to chlorine carryover [89,90,91]. Henthorne [92] mentioned numerous pilot studies in the United States and globally has had similar experiences. The following are examples of RO membrane damages due to chlorine carryover. First, Henthorne and Quigley [89] describe SWRO membrane damage caused by chlorine from the LP membrane filtration CEB cycles and a dead spot in the pipe in which chlorine is accumulated. Then, residual chlorine was subsequently fed into the RO system. Thus, the SBS dosage was increased to 2 ppm, and the frequency of the chlorine CEB was reduced to further remedy this problem. The following case is the Brownsville seawater desalination demonstration facility that started up in February 2007 [90,91]. It was observed that salt rejection of one train was not consistent with typical SWRO permeate and permeate conductivity approached 1 mS/cm. The autopsy results were that the RO membrane surface was halogenated from oxidation. The cause of chlorine breakthrough was identified as the failure of a solenoid valve controlling the chlorine injection timing. In addition to fixing the solenoid, the membrane pretreatment flushing procedures were modified to account for an extended flush time.

As for the pilot plant failures and chlorine carryover issues in the CEB process, Henthorne [92] speculates that even though the RO industry recognizes the need to prevent RO damage from chlorine attack, it was not considered a potential problem with the LP membrane filtration units at that time. Further, this author also noted that the RO membrane damage originating from the CEB should be one of the most significant issues associated with using membrane filtration as the pretreatment to RO.

Several countermeasures were proposed for smaller systems and pilot tests to prevent RO membrane oxidation. Continuous dosing of a small amount of SMBS might be an option, as described in Section 3.1 and Section 3.4 [93]. However, for large desalination plants, more sophisticated control might be necessary. The first step is to better understand the exact effluent characteristics from the LP membranes during backwash and CEB. One pilot test found that trace amounts of chlorinated solution were generated, even after 25 min of service following a flush [94]. A similar result was observed in another pilot test conducted at Marbella, Spain [95]. A certain type of UF module is cleaned with a 200 ppm NaOCl solution (Maintenance Cleanings: MC1) once or twice per day. Even when the UF is thoroughly rinsed, it was found that the filtrate has higher than 350 mV of ORP up to two hours after filtration was resumed. Thus, when ORP is higher than 350 mV, an SBS dosing system was made available to avoid membrane oxidation. Given that continuous dosing of SBS may promote biofouling and an excess of SBS may lead to membrane oxidation, SBS is dosed at 1 ppm only during the two hours after every MC1. The rest of the time, no SBS is dosed. As a result, no ORP reading exceeds 350 mV, considerably reducing the chemical consumption. As demonstrated here, both enough rinse-down after cleanings and good operational controls are of critical importance [88,96].

Suárez et al. [97,98] shared their experiences on avoiding the RO membrane oxidation in the IMS plants. Maspalomas-I Desalination Plant, located on Gran Canaria, Spain, has an original capacity of 14,500 m^3^/d. In addition to plant expansion, the existing conventional pretreatment was substituted by UF technology. When designing and operating the UF plant, special care was taken to address the issues of chlorine carryover to RO. First, a thorough rinse via backwash is carried out in the UF trains after exposure to chlorine. Moreover, as an extra safety measure, once any UF train comes back to filtration after cleaning with chlorine, the initial UF filtrate volume produced is sent for a few minutes to drain through an out-of-spec line until the residual chlorine is below 0.20 ppm. In addition, SBS is dosed temporarily at the UF product tank inlet.

Busch et al. [88] suggested key measures each plant should take in detail. Potential protocols for improved inhibition of oxidative damage could consist out of various elements, including leakage monitoring, improved CEB practices, redox control, and SMBS safety, as well as event dosing.

### 3.5. Other Disinfectants Removal

As mentioned, biofouling is one of the critical issues in RO operation. Chlorine is the most efficient and economical chemical to disinfect RO feedwater to prevent biofouling. However, disinfection by-product (DPBs) formation, such as trihalomethanes (THMs) and the risk of RO membrane oxidation, are of concern. Thus other types of disinfectants have been investigated and used [36]. Those include combined halogen disinfectants (chloramine and chlorosulfamate, etc.) [99,100], weak oxidants (chlorine dioxide and peracetic acid, etc.) [101], and nonoxidative biocides (2,2-dibromo-3-nitrilopropionamide (DBNPA) and Isothiazolones) [102].

Among those disinfectants or sanitizing agents, chloramine is the most commonly used in RO or contained in feed water (municipal water) [103]. Applegate et al. [104] proposed using chloramine due to bacterial aftergrowth in the chlorination–dechlorination process. Chloramine did not degrade humic acid and assimilable organic carbon (AOC) was not generated. In addition, significantly less aftergrowth was observed in the chloramine process. Based on those findings, the chloramine disinfection process was first applied to a seawater desalination plant on the island of Java in Indonesia [105]. The other benefit of chloramine is less THM formation compared with chlorination, as CA membranes show low THM rejection or even negative rejection [106,107]. For that reason, Tanaka et al. [108] proposed to use chloramine for seawater disinfection from the point of THM formation. It was confirmed that there are no THMs in chloramine-disinfected seawater. In cellulose triacetate (CTA) hollow fiber RO, chloramine-containing feed water can be continuously supplied. It was observed that chloramine disinfects microorganisms in seawater just as effectively as chlorine. Another positive result was derived from troubleshooting efforts of CA RO membrane oxidation in the Yuma Desalting Plant [109]. The Yuma Desalting Plant was built to help accomplish salinity control of Colorado River water. Premature loss of salt rejection by cellulose acetate membranes was experienced during test operations. Later on, this membrane degradation was attributed to a catalyzed (by traces of iron and other heavy metals) hypochlorite oxidation [110]. It was found that converting free chlorine to chloramines by injecting ammonia in the RO feed water could solve the problem. The actual plant started in March 1992 with the chloramine conversion method [109].

DBNPA is a new type of disinfectant for RO, which is classified as a non-oxidative biocide. DBNPA has been used for various water treatments, e.g., cooling water, pulp and paper, and enhanced oil recovery, etc. [111]. For example, Durham [112] introduced two cases in which DBNPA was intermittently injected every week or two weeks. The subsequent earlier trial is observed in a makeup system at Huntington Beach Generating Station. The makeup water plant was built in April 1993 based on an RO-EDI hybrid process. In an effort to minimize the need for chemical cleaning and prevent biofouling problems, the plant decided to dose DBNPA intermittently (20 ppm of DBNPA for 60 min) [113].

As aforementioned, THM formation during chlorination is an issue for permeate water quality. Thus, Tanaka et al. [114] also evaluated chlorine dioxide (ClO_2_) as an alternative disinfectant for seawater desalination. As a result, it was confirmed that there was no THM in the chlorine dioxide-disinfected seawater. Furthermore, oxidative membrane degradation was not observed for about one year, and RO performances were stable.

Although those disinfectants are considered compatible or partially compatible with TFC polyamide RO membranes, it is known that the RO membranes are degraded under specific conditions. Thus, some types of disinfectants may have to be removed from feed waters prior to entering the RO. Table 6 summarizes oxidant removal needs for three disinfectants: chloramine, chlorine dioxide, and DBNPA.

#### 3.5.1. Chloramine

Although chloramine is a weak oxidant, it is not compatible with the aramid hollow fiber RO membrane. Thus, it was requested that chloramine is neutralized with SBS [104,105]. On the other hand, the TFC RO membranes generally have some tolerance for chloramine that depends on membrane types [23,115,116,117]. The better chloramine tolerance implies that dechlorination may not be required. In this regard, chloramine has been successfully applied for municipal wastewater treatment [118,119,120,121]. However, it is reported that membrane degradation occurs at specific conditions and for other feed waters [36]. One possible cause is originated from a chloramine generation method itself. As chloramines are formed by adding ammonia to chlorine, free chlorine may be present in feed [23,115]. The next factor inducing membrane degradation is the presence of bromide ion in feed waters. It is known that exposure to ammonium salts with chlorinated seawater forms bromamines. Bromamine is a more potent oxidant than chloramine, damaging the downstream RO membranes [34,36,122]. Intensive research works were conducted during the Ocean Water Desalination pilot and demonstration projects at West Basin Municipal Water District (Carson, CA, USA) [123,124,125,126]. Sharma et al. [124] clarified the presence of bromamines by measuring UV absorbance when injecting ammonium salt into pre-chlorinated seawater. The authors also evaluated the behavior of preformed chloramine. It was found that the preformed chloramine was stable in seawater for less than an hour. However, the preformed chloramine is gradually converted to bromochloramine. This transformed bromochloramine may induce another membrane oxidation risk. The demonstration project report [126] mentions that when the RO system was shut down for longer than a few hours, membranes were chemically damaged by a strong oxidant formed by the reaction of chloramines with bromides present in seawater trapped in the annular space of the pressure vessels. Such shutdown events would require flushing with de-chloraminated RO permeate water. In Table 7, three scenarios of bromamines formation are summarized. The preformed chloramine dosing is an idea to prevent membrane oxidation. However, as observed from the reaction scheme, the bromochloramine formation is dependent on pH. It is mentioned that at 25 °C and salinity of 35,000 mg/L, the half-life of the reaction is 8 h at pH 8.0 but only 45 min at pH 7 [104]. Valentine [127] produced bromochloramine by adding bromide to solutions of NH_2_Cl at pH 6.5. Soon after adding bromide, NHBrCl was quickly generated. Thus when SWRO plants need to operate with lower pH to prevent carbonate scaling, membrane oxidation with bromochloramine may still be a risk even though the preformed chloramine is dosed.

The third factor of membrane oxidation with chloramine is the presence of heavy metals. There are several reports that heavy metals (Fe(II), Fe(III), Al, and Cu, etc.) catalyzed membrane oxidation [128,129,130,131,132]. Gabelich et al. [129] indicated that the formation of an amidogen radical (∙NH_2_) during NH_2_Cl decomposition with Fe(II) led to the reduction of the activation energy for the chlorination reaction to proceed using NH_2_Cl. Fu et al. [133] investigated the mechanism of Cu(II)-catalyzed monochloramine decomposition. Electron spin resonance (ESR) results demonstrated that the hydroxyl radical (∙OH) and amidogen radical (∙NH_2_) were generated in the reaction between monochloramine and Cu(II). Upon formation, ∙OH could maintain a strong intensity longer than ∙NH_2_ in the reaction solution. In this NH_2_Cl–Cu(II) system, the authors also measured the effect of the solution pH. The results indicate that the radical intensity significantly decreased with the increase of pH. More than 80% of the radical intermediates disappeared as the solution pH was raised from 5.8 to 7.9. Based on these findings, one may consider the effect of hydroxyl radical and amidogen radical as causes of membrane oxidation by chloramine. This result is consistent with the report by Cran et al. [132]. Degradation of RO membranes was evaluated in the presence of heavy metals (Al^3+^, Fe^2+^, Al^3+^/Fe^2+^, and Cu^2+^). It was observed that the stability of chloramine solutions in the presence of metal ions decreased significantly with Cu^2+^ and a combination of Al^3+^/Fe^2+^. The presence of Cu^2+^ with chloramine significantly accelerated the reduction of the amide (II) absorbance (1540 cm^−1^) of the polyamide RO membrane. As for remediation methods relating to membrane oxidation, Gabelich et al. [134] reported the effect of citric acid as a chelating agent for Al^3+^. When a chelating agent (citric acid, 5 mg/L) was added to the RO feed (1.5–2.5 mg/L chloramines present), the loss in productivity and selectivity was arrested. In this case, citric acid may act as both a radical scavenger and a chelating agent [135]. It is known that some types of antiscalants have a role in chelating action. This information might be a hint to understand successful cases in some surface and ground water treatment plants where chloramine disinfection has been applied together with antiscalants.

In this section, the mechanism of membrane degradation by chloramine was discussed. It is becoming clearer how to prevent membrane oxidation. However, it appears that there are still unexamined and unsolved issues. Thus, it might be better to consider eliminating chloramine prior to reaching RO membranes, except for in municipal wastewater applications [36,136].

Another area of a need for chloramine removal is from RO brine. Due to concern about the environmental impact of discharge water, chloramine removal may be requested from local municipalities [137]. For example, the Murrumba Downs Advanced Water Treatment Plant in Queensland, Australia, implemented dechloramination of RO brine before discharge [138]. Dechloramination was achieved by SBS injection. The treated RO concentrate is captured in a storage tank and then discharged with the effluent from the wastewater treatment plant.

The dechloramination methods are quite similar to dechlorination. Dechloramination is typically accomplished by either SMBS/SBS or AC in RO [36]. The reactions of SBS and SMBS with monochloramine are as follows:NaHSO_3_ + NH_2_Cl + H_2_O → NaHSO_4_ + NH_4_Cl(10)
Na_2_S_2_O_5_ + 2NH_2_Cl + 3H_2_O → 2NaHSO_4_ + 2NH_4_Cl(11)

Stoichiometrically 2.0 mg of SBS or 1.85 mg of SMBS removes 1.0 mg of monochloramine. It is said that the reaction for SBS is rapid and as fast as the neutralization of chlorine [104]. Basu and Souza [39] measured the dechloramination rate with SBS and compared it with dechlorination. The removal of monochloramine using a 3× stoichiometric dosage of SBS occurred quickly, with a completion time of approximately 32 s compared to 42 s for the control free chlorine solution. However, Ekkad and Huber [139] reported contradicting reaction times. In their report, the following calculated reaction times were indicated for chlorine and chloramine (1 µM concentrations):Free chlorine (pH < 11.0): 13 ms;Free chlorine (pH > 11.0): 4.3 s;Monochloramine (pH 4.0): 1.8 s;Monochloramine (pH 8.0): 2.0 min.

As observed, the reaction time is pH dependent. Dechloramination reactions are rapid at low pH 4.0. But at slightly alkaline pH, the reaction of sulfite with chloramine is much slower (2.0 min). Relating to this phenomenon, Comb [140,141] reported case studies where SBS was added for dechloramination upstream of polyamide (PA) membrane RO systems. In the cases of higher pH 8.5, SBS proved to be ineffective at reducing chloramines and then maintaining an entirely reduced state. Thus, operating at higher pH resulted in membrane oxidation, as evidenced by higher salt passage. However, when the feed pH is acidic, SBS effectively reduced 4 ppm of chloramines to the point where PA membrane oxidation is avoided for more than 3 years. Thus, the author concludes that pH most likely plays a vital role in reacting chloramines and bisulfite.

Although there exist some anomalous points for dechloramination chemistry by SBS, the following measures should be taken into consideration: taking enough contact time with complete mixing, adjusting feed pH, and monitoring the ORP readings.

#### 3.5.2. Chlorine Dioxide (ClO_2_)

Although ClO_2_ is considered less oxidative in nature and applicable to polyamide RO membranes, there are some conflicts about the compatibility of chlorine dioxide and polyamide membranes. This ambiguity might be a stumbling block to applying ClO_2_ to RO. Kucera [36] summarizes limitations and precautions using ClO_2_ to RO. Potential risks and issues using chlorine dioxide are as follows:A risk of containing residual chlorine (preparation issue) [68];Strong oxidant generation when bromide ion is contained in feed waters (e.g., seawater) [142,143];Membrane oxidation at higher pH (e.g., >pH 8.0) [144,145,146];Effects of heavy metals catalyzing the membrane oxidation [147];Free chlorine/bromine generation in the presence of natural organic matter (NOM) [148].

It seems ClO_2_ itself has a less oxidative capability for RO. However, additional factors may accelerate membrane degradation, such as pH, bromide ion, and NOM, etc. It is generally said that ClO_2_ does not oxidize bromide to bromine or hypobromite [149]. However, some reports demonstrate that bromide ion contributes to RO membrane deterioration, as observed in the chloramine cases [142,143]. Sandín et al. [142] measured the effect of feed compositions: pure water, NaCl solution, and seawater. They observed a noticeable salt rejection decline when ClO_2_ was present in seawater. The bromine atom was detected from the seawater ClO_2_-treated membrane sample by the X-ray photoelectron spectroscopy (XPS) analysis. They speculated that the behavior difference from that observed in pure water and NaCl solution is related to the bromide content of seawater. Mizuta [143] also mentioned that ClO_2_ oxidized bromide when the bromide concentration exceeded that of ClO_2_.

The next factor affecting RO membrane performance is feed pH. It has been recognized that higher pH exposure results in a more significant loss of salt rejection [36]. Alayemieka and Lee [144] evaluated the effect of ClO_2_ on RO membrane characteristics at different pH. They observed that the salt rejection was apparently decreased after 100 ppm·h ClO_2_ contact (20 ppm × 5 h) at pH 9.0. Further, the membrane surface composition immersed at pH 9.0 was considerably different from those treated at neutral or acidic conditions. Kim [145] conducted similar experiments with wider pH ranges: pH 4.0, 7.0, 10.0, and 12.0. At higher pH conditions, it was confirmed that ClO_2_ heavily damaged RO/NF membranes. The scanning electron microscopy (SEM) analysis observed that the thin film polyamide layer almost disappeared for a sample treated at 100 ppm·h ClO_2_ contact (100 ppm × 1 h) at pH 12.0. As observed by Alayemieka and Lee, the chlorine content of the polyamide layer treated with pH 10.0 and 12.0 is less than those for pH 4.0 and 7.0 samples. It was also confirmed that despite very low contact (5 ppm·h, ClO_2_) at pH 12.0, the polyamide NF membrane was chemically attacked. As less chlorine atoms were detected at pH 10.0 and 12.0, a hypochlorite (OCl^−^) attack may not be considered a cause of membrane degradation. Regarding the membrane degradation at high pH, Kim postulated the role of hydroxyl radical (∙OH) and conducted an additional experiment using benzoic acid as an OH radical scavenger. However, in this method, ∙OH radical was not detected.
ClO_2_ + OH^−^ → ClO_2_^−^ + ∙OH(12)

In this regard, Marcon et al. [146] utilized electron paramagnetic resonance (EPR) spectroscopy based on the spin-trapping technique to identify the mechanism of ClO_2_ decomposition in an alkaline medium. They confirmed the presence of hydroxyl radicals (∙OH) at alkaline pH with this method. They speculated that the generation of ∙OH could be one reason for cellulose degradation by ClO_2_ at alkaline pH. The ∙OH radical formation could well explain the intense attack on polyamide membranes at higher pH. However, as there are still unclear points about the mechanism of membrane degradation, further studies might be needed.

The last unknown factor is the effect of NOM contained in feed waters. It is said that free available chlorine (FAC) is formed during the oxidation of organic compounds with ClO_2_. Hupperich et al. [148] evaluated the effect of NOM and some model compounds, including phenols and olefins. When treating the Suwannee River NOM solution (5 mg/L DOC) with ClO_2_, it was observed that a fair amount of free available chlorine (22%) is formed in addition to the following products, chlorite (63%), chloride (8%), and chlorate (5%). Although there is no systematic analysis of the effect of NOM on RO systems, great care may be required when dealing with higher TOC waters.

Up to this point, several potential risks using ClO_2_ as a disinfectant to RO were reviewed. Although there are some clear benefits to using ClO_2_, one should be cautious about using ClO_2_ for continuous dosing or sanitization to RO until further investigation is conducted. Otherwise, it is recommended to remove all ClO_2_ prior to RO [36]. For example, the Tampa Bay Seawater Desalination plant implemented a unique double disinfection process in which ClO_2_ is injected into the feed intake to address issues of green mussel growth and THM formation [150]. Chlorine is dosed as the process disinfectant. SBS is used to remove ClO_2_ and residual chlorine.

Another issue using ClO_2_ to RO is the formation of DBPs, the chlorite ion (ClO_2_^−^), and chlorate ion (ClO_3_^−^). It is known that soon after ClO_2_ is added to water, approximately 50–70% of ClO_2_ is immediately converted to ClO_2_^−^ and ClO_3_^−^ [114,151,152]. In Japan, chlorate is regulated at a concentration of 0.6 mg/L for drinking water. The World Health Organization (WHO) recommends a chlorite and chlorate limit of 0.7 mg/L each. As both ClO_2_^−^ and ClO_3_^−^ could be well removed by RO, the RO permeate quality may not be concerning. However, when discharging the RO brine containing ClO_2_, ClO_2_^−^, and ClO_3_^−^ to an environmentally sensitive area, such as marine reserves, ClO_2_ and its DBPs may have to be removed.

Regarding the effect of ClO_2_^−^ and ClO_3_^−^ ions on the RO membrane, Ferrero et al. [153] conducted laboratory tests to determine the resistance of various polyamide RO membranes on water solutions containing 100 mg/L of ClO_2_^−^ or ClO_3_^−^. In the tests, the membranes were also characterized by FTIR-ATR after the treatment. There was no sign of a chemical attack for the polyamide active layer. The membrane performance did not change after 35,000 ppm·h (100 ppm × 350 h) contact. Thus, when ClO_2_ needs to be removed from waters containing ClO_2_^−^ or ClO_3_^−^ ions to protect RO, the ClO_2_ removal itself may be of more concern.

Chlorine dioxide removal can be done by sulfites, thiosulfate, activated carbon, and ferrous salts. The reactions of sodium sulfite and sodium thiosulfate with ClO_2_ are as follows [36]:5Na_2_SO_3_ + 2ClO_2_ + H_2_O → 5Na_2_SO_4_ + 2HCl(13)
5Na_2_S_2_O_3_ + 8ClO_2_ + 9H_2_O → 10Na_2_HSO_4_ + 8HCl(14)

Based on these reactions, theoretical dosages for different sulfites and thiosulfate are summarized in Table 8.

The reaction of sulfites with ClO_2_ is rapid. Ekkad and Huber [139] reported that the reaction time of sulfites are 0.1 s at pH 9.0 and 11.0, respectively, and are comparable to that of chlorine. Suzuki and Gordon [154] measured the chlorine dioxide-S(IV) reaction with a slight excess of S(IV). They observed that the reactions were relatively rapid such that they were finished within 5 s. The same results are observed in the unexamined patent publication [155] (JPH0929075A). A total of 0.6–1.0 mg/L ClO_2_ is dosed to seawater and then supplied to cellulose acetate RO. As a result, 0.1–0.3 mg/L of residual ClO_2_ is detected in the RO permeates. By injecting 0.5–0.8 mg/L of SBS into the permeate water, the residual ClO_2_ is eliminated after 5 min of contact. Based on these findings, it seems that a slight excess amount of SBS is enough to remove residual ClO_2_ from RO feed waters.

However, a new problem emerges when ClO_2_^−^ ion needs to be removed from feed water and RO brine by SBS. ClO_2_^−^ can be reduced to chloride by sulfite ion, and this reaction is efficient when the pH is between 5.0 and 6.5. The reaction slows markedly at pH above 7.0 and is too slow for water treatment at very high pH values [114,151,156]. With a 10-fold excess of the sulfite ion, and a ClO_2_^−^ residual of 0.5–7.0 mg/L, complete removal of the ClO_2_^−^ occurred in less than 1 min at pH values less than 5.0. At pH 6.5, less than 15 min was required. Thus, the excess amount of SBS should be added in order to complete the reaction within 5–10 min. This excess SBS dosing creates other critical problems: strong oxidant generation from SBS and increase of ClO_3_^−^ concentration. As described later, under specific conditions, such as heavy metal (Cu and Co) presence in feed waters, a strong oxidant is generated when an excessive amount of SBS exists. Tanaka et al. [114] observed that SBS generated oxidizing agents with a 5-min contact period when 10 mg/L of SBS was added to RO brine water, where ClO_2_ and ClO_2_^−^ concentration is about 0.1 mg/L and 0.6 mg/L, respectively. To address this issue, they proposed to use sodium thiosulfate. Sodium thiosulfate reduced chlorite ion to chloride at the neutral pH (6.7–7.2) in RO brine water without forming oxidizing agents. Doñaque et al. [157] investigated the effect of using ClO_2_ for seawater desalination treatment and on the DBPs. This study evaluated ClO_2_ and ClO_2_^−^ removal capability with SBS for seawater and 100 μg/L of Cu(II)-spiked seawater. In the case of seawater, it was observed that both ClO_2_ and ClO_2_^−^ concentrations were increased. Concentration of ClO_2_ was increased from 0.4 mg/L to 0.97 mg/L after dosing 10 mg/L of SBS. This result does not seem to match the previous data by Tanaka et al. [114]. In their report, an unknown oxidant was generated rather than ClO_2_, and ClO_2_ concentration was not increased. This might come from differences in the ClO_2_ concentration analysis method. As for the effect of Cu(II) ion, similar results were observed in which both ClO_2_ and ClO_2_^−^ concentrations were increased. However, compared with seawater, a noticeable increase of ClO_2_^−^ ion concentration was observed. Furthermore, the ORP value was increased to 752 mV even though 10 mg/L of SBS was added. The author postulated that the Cu(II) ion catalytically oxidizes the bisulfite ions into persulfate or peroxodisulfate anions, which simultaneously regenerate ClO_2_ and increase ClO_2_^−^ ion concentration from the high concentration of chlorides in seawater. [157]. Thus, care must be taken not to make the RO membrane deteriorate when adding excess SBS to remove ClO_2_ and ClO_2_^−^, especially to RO feed. Careful ORP monitoring is essential for both feed and brine in this situation.

Another issue is an increase of ClO_3_^−^ concentration when adding excessive SBS. This is the case of a pilot test conducted at the Evansville Water and Sewer Utility. Griese et al. [158] reported the bench-scale test results in which excessive sulfur dioxide (25 mg/L of SO_2_) was applied to treat waters with a variety of ClO_2_ dosages. It was observed that oxygenated water supplies containing ClO_2_^−^ formed ClO_3_^−^ when treated with SO_2_. Although complete reduction of residual ClO_2_ and ClO_2_^−^ was achieved after 30 min of contact time, a marked increase in ClO_3_^−^ concentration was consistently observed. The same result was observed in a lab test using SMBS. This contradicts the previous results obtained for waters with the absence of oxygen. In the absence of oxygen, a chlorite removal reaction with sulfite followed the reaction to produce sulfate and chloride, and no ClO_3_^−^ is formed [151]:2SO_3_^2−^ + ClO_2_^−^ → 2SO_4_^−^ + Cl^−^(15)

Griese et al. [158] mentioned that these reactions are complicated for oxygenated waters. Several different pathways result in the reduction of ClO_2_^−^ to chloride ion and the formation of ClO_3_^−^ as an unwanted inorganic by-product. They pointed out that the potential benefits associated with the use of SO_2_/SO_3_^2−^ for the reduction of DBPs appear to be severely limited.

Again, this phenomenon could be partially explained by autoxidation reactions of sulfite in the presence of oxygen in which strong oxidants and radicals are generated, as discussed in Section 8.

In summary, to remove ClO_2_ and ClO_2_^−^ within an acceptable reaction time at neutral pH, an excessive amount of SBS may have to be injected. However, this results in risks generating strong oxidants and increasing ClO_3_^−^ ion. To avoid those risks, using thiosulfate instead of SBS might be an option. The other option is to use ferrous salts injection for RO brine or prior to a media filter or LP membranes [159]. When treating RO brine containing 0.3 mg/L of ClO_2_ and 0.9 mg/L of ClO_2_^−^ with 10 mg/L of ferrous ammonium sulfate, both ClO_2_ and ClO_2_^−^ are removed without forming the ClO_3_^−^ ion [155]. Doñaque et al. [157] reported the ion Fe(II) is oxidized to Fe(III) in a fast reaction that results in eliminating Cl_2_, ClO_2_, and ClO_2_^−^ and producing FeCl_3_, which can act as an effective coagulant.

#### 3.5.3. 2,2-Dibromo-3-nitrilopropionamide (DBNPA)

DBNPA is a non-oxidative biocide that can be used for RO continuously or intermittently. Furthermore, a high concentration of DBNPA could be used for RO system sanitization after CIP. However, when discharging RO brine or sanitizing effluent to environmentally sensitive areas, DBNPA may have to be removed. Elimination of DBNPA is accomplished by dosing SBS [160,161]. Reduction by SBS yields cyanoacetic acid and two equivalents of bromide ions [162].
N≡C-CBr_2_-CONH_2_ + 2NaHSO_3_ + 2H_2_O → N≡C-CH_2_-CONH_2_ + 2H_2_SO_4_ + 2NaBr (DBNPA)                      (Cyanoacetamide) (16)

Boorsma et al. [163] reported the IMS surface water treatment plant in Klazienaveen, the Netherlands. They reported that intermittent dosing of DBNPA successfully controlled biofouling. DBNPA was neutralized before the discharge in the wastewater pond and subsequent release into the surface water. SBS was applied for neutralization, and ORP was used to monitor adequate neutralization.

## 4. Preservative for New RO Elements and Storage in Plant Shutdown

After dechlorination, membrane preservation is the second most-used application of sulfites in the RO process. Preservation of the RO elements is essential in two areas: the preservation of new RO elements and storage during plant shutdown. First of all, the new RO elements are shipped with a preserving solution to prevent biofouling. In the past, a 0.3–1.0 wt% solution of formaldehyde was commonly used as a shipping solution for CA RO elements [164,165,166]. However, due to a concern about health effects as a potential occupational carcinogen, formaldehyde has been obsoleted in the RO process.

When the polyamide hollow fiber RO was developed, the RO modules were treated with a 0.25 wt% SMBS and 18 wt% glycerine solution prior to shipment [167]. By following this procedure, TFC polyamide spiral RO elements were shipped with a solution of 20 wt% glycerine and 1.0 wt% SBS (food grade) [168,169]. This solution also protects from freeze damage. Later on, glycerine was switched to propylene glycol, and then propylene glycol was eliminated from the shipping solution. The role of SMBS is a biostatic agent to prevent bacterial growth within the RO elements. In addition, SMBS acts as the oxygen scavenger. As the polyether composite RO (PEC-1000) is less tolerable to oxygen, sulfites and an iron-based oxygen scavenger were evaluated to protect the RO membrane [170]. It was reported that 0.5% SMBS and deoxidizer packets kept the oxygen level low enough in the RO element without changing the performance for one year. It is interesting that the iron-based deoxidizer packets evaluated at that time have presently been implemented to a certain type of RO element [171].

Most spiral-wound RO elements are currently preserved with a 0.5–1.0% SMBS solution in oxygen barrier plastic bags [34]. In addition, a certain type of RO element is preserved in a buffered SMBS solution using sodium citrate to mitigate pH changes. For storage lasting longer than six months, preserved elements should be visually inspected for biological growth and periodically examined every three months after that. If the preservation solution appears to be murky, the elements should be re-preserved and vacuum-sealed. Another method for checking the integrity of the preservative is through pH measurements. The bisulfite in the preservative can oxidize into sulfuric acid, which will cause the pH to drop. If the pH of the preservative drops below 3, the elements must be re-preserved [172].

Next, the storage application is for the plant shutdown case. When the RO system needs to be shut down for longer than 48 h, necessary measures must be taken to prevent microorganism growth. Membrane suppliers suggest such measures depending on storage periods: short-term storage, 1–2 weeks or less, for example, and long-term storage, more than 1–2 weeks [173,174,175]. For short-term storage, flushing with RO permeate or filtered feed water is generally recommended. Regarding long-term storage, it is recommended that the RO elements be stored within entire RO racks or oxygen barrier plastic bags with a 0.5–1.0% SMBS solution. One membrane supplier suggests using a lower concentration of SBS solution, i.e., 500 to 1000 mg/L (maximum) [176].

As mentioned, a 0.5–1.0% SMBS solution is now commonly used as a long-term preservative. Until now, several tests have been made to examine the storage conditions and compare an SMBS solution with other preservatives. Here, brief chronological highlights will be shown. As mentioned, for the polyamide hollow fiber RO, the use of 0.25% of SMBS was recommended. Furthermore, an addition of 18 wt% glycerine was essential to prevent biological growth [167]. Larson et al. [177] reported that the best FT-30 RO membrane storage procedure is to store the element in a 0.1% aqueous SBS after various storage tests. However, later, Petersen et al. [178] reported that SBS or SMBS, used at 0.5% in water, appear preferable for shelf storage or prolonged “down” periods. Henthorne et al. [179] reported the comparison test results as a part of a cooperative research program between the United States Department of Interior, Bureau of Reclamation (BR), and the Kingdom of Saudi Arabia, Saline Water Conversion Corporation (SWCC). In this cooperative research, three types of biocides were evaluated; Minncare™ (a peracetic acid solution), Bronopol™, and SBS. The three biocides chosen for the testing were based on a screening evaluation of 13 potential biocides conducted for the Yuma Desalting Plant [166,180]. A total of 3% SBS was evaluated at the BR test with keeping the solution pH at approximately 5.5. The SBS concentration utilized in SWCC was 400 mg/L. The SBS solution was replaced every two weeks and pH adjusted to 4 +/− 0.2. In the SWCC test, no salt rejection decline was observed after 36 months of storage. On the other hand, a slight increase of the normalized permeate flow (NPF) was observed. The cumulative testing indicated that TFC SWRO membranes stored in the three tested biocides respond in the following order of acceptability of biocides: SBS >> Bronopol™ >> Minncare™.

When storing the RO elements with the SMBS solution, the following two points should be noted: the decrease in pH of the SMBS preservative solution and the heavy metal fouling of the membrane surface. As shown in Equation (4), when SBS in the preservative contacts with oxygen intruded into RO racks or storage plastic bags, SBS is oxidized to sulfuric acid, which will cause the pH to drop. In this regard, several tests were conducted to elucidate the effect of pH. The Naval Facilities Engineering Service Center (NFESC) conducted a three-year test program to evaluate the effectiveness of seven preservatives for TFC SWRO membranes. A 1% SBS and 18% propylene glycol solution was also evaluated as a generic storage solution. It was reported [175] that the SBS-based preservative was particularly detrimental to salt rejection performance. The preserved elements had a drop in normalized salt rejection greater than 0.30%, while the control group declined about 0.25%. This result contradicts the previous results on membrane compatibility. In the NFESC test, the average SBS solution pH was 3.17, lower than the BR and SWCC tests. Although the authors did not touch on the pH effect, this might be a potential cause of the salt rejection decline. To avoid the pH changes during storage, Ventresque et al. [181] decided to put the membranes into bags and preserve them with SBS, which was added phosphate buffer to stabilize the pH, thus avoiding frequent refills of preservative solution. After eight months of storage, membranes are fitted again in the pressure vessels, rinsed, and returned to service. No degradation of the permeability or retention was observed.

Tu et al. [182] evaluated three preservatives, namely formaldehyde, SMBS, and DBNPA. SMBS at 5% and formaldehyde preservative solutions adjusted to either pH 3.0 or 7.0 were used for a 14 days storage test. When the pH of the SMBS and formaldehyde solutions was reduced to 3.0, prominent boron and sodium rejection declines were observed. The authors suggest a near-neutral pH (i.e., pH 7.0) is necessary to avoid significant negative impacts on membrane performance using SMBS. In addition, some changes in the membrane surface properties (zeta-potential and FT-IR absorbance) were also observed. Apart from the RO membrane degradation by SBS, Ventresque et al. [183] reported an adverse effect to pressure vessels at low pH. They found that even though pH was above 3.0 in the water body during preservation, lower pH had been induced in the air trapped within the pressure vessels above the SBS solution. Acid attack weakens the resin and the glass fiber, which then cracks easily under low stress.

Thus far, the effect of pH of the SBS solution on the RO performance during storage was reviewed. It is confirmed that SBS solution pH less than 3.0 has a phenomenologically negative impact on RO. However, it seems that the mechanism and cause of deterioration of membrane performance are still not clear, and it may need further tests to know which pH level is safe for long-term storage from a practical point of view.

The next factor to be considered is the effect of heavy metal fouling on the RO membrane surface during storage. It is reported [184] that the rejection performance deteriorated when heavy metal-fouled RO membranes were stored in an SBS solution. The inventors found that in a system in which heavy metals, such as copper and chromium, are present in RO membranes, SBS generates an oxidizing substance that results in membrane degradation. Furthermore, it was found that the deterioration of membrane performance could be suppressed by adding a small amount of a chelating agent. Farooque et al. [185] reported that the polyamide hollow fiber RO encountered the problem of high permeate conductivity in some of the BWRO membranes, which were preserved in SBS solution for about 23 days due to plant shutdown for annual maintenance. From the SEM-EDX analysis, a high level of Fe and Cr was detected from the fiber surface. Furthermore, oxidative degradation was confirmed by measuring the polyamide intrinsic viscosity. However, the authors suspected that the membrane could have been accidentally exposed to chlorine.

Ventresque et al. [183] summarized the storage methods by SBS as follows:Clean membrane before applying SBS.Immerse membranes in the preservation solution directly in the pressure vessels.Vent the air from the system and isolate the system.Check pH during preservation to monitor the degradation of the preservation solution.Change preservation solution if pH is below 3.0.Change preservation solution every 30 days if the temperature is below 27 °C and 15 days if the temperature is above 27 °C.

Cleaning should be an essential step to remove heavy metals and prevent membrane degradation. Venting air and isolating the system can minimize the SBS oxidation and decrease pH. A regular pH check is imperative to confirm the SBS storage solution conditions. Additionally, adding chelating agents into the preservative might be an option to prevent membrane oxidation [174]. When using sodium citrate, this chemical may have the triple roles acting as a buffer, a chelating agent, and a radical scavenger.

As mentioned, SMBS is now the most commonly utilized preservative in RO plants during shutdown. It is a cheap and efficient preservative, but its tendency to oxidize easily has several drawbacks: a need for regular pH checks, isolation of the RO system from the air, and an odor issue due to SO_2_ gas release, etc. [186]. Therefore, studies have been made to apply non-oxidative biocides as membrane preservatives. Majamaa et al. [186] conducted long-term preservation trials by using three different non-oxidative biocides: DBNPA, 5-Chloro-2-methyl-4-isothiazolin-3-one (CMIT)/2-Methyl-4-isothiazolin-3-one (MIT) CMIT/MIT, 2-Octyl-2H-isothiazol-3-one (OIT) as well as SMBS as a reference chemical. It was demonstrated that the biocides can be equivalent preservatives to SMBS and that the application is economically feasible.

Regarding DBNPA, Tu et al. [182] also evaluated its compatibility with TFC RO membranes. They found that salt rejections (boron and Na ion) declined at neutral pH (pH 7.0). However, it is noted that the concentration of DBNPA (1%) used is much higher than that (60 ppm) used by Majamaa et al.

The non-oxidative biocide, CMIT/MIT, was applied to the Camp de Tarragona-Vilaseca Water Reclamation Plant, Spain, for nearly ten months of storage [187,188]. Prior to applying the CMIT/MIT, a complete CIP was conducted with caustic and acid cleaning solutions. Then, the biocide was added by recirculating feed water containing 15 mg/L of CMIT/MIT through the RO pressure vessels. After six months, RO pressure vessels were drained, and a new biocide solution was put in place. An evaluation of performance before and after shutdown demonstrated that membrane performance after the extended shutdown was similar to new performance during the commissioning period [189]. The isothiazoline-based biocide was also applied to the Barcelona SWRO Plant [190].

## 5. Deoxygenation

In boiler-water treatment and oilfield production, sulfites are used as an oxygen scavenger. In a seawater distillation process, such as multi-stage flash evaporation (MSF), SBS is the most used to remove residual oxygen after mechanical deaeration [191]. In some RO applications, deoxygenation is necessary. There are two main applications applying sulfites to remove oxygen from feed waters.

The first application is to protect an RO membrane from degradation by oxygen. In the late 1970s, a new TFC membrane, designated PEC-1000, was developed [192,193]. The PEC-100 made it possible to produce potable water from seawater in a single stage with a high recovery operation. However, as the PEC-1000 membrane had no chlorine and oxygen resistivity, it was necessary to eliminate the dissolved oxygen (DO) by SBS [26,194,195]. DO was requested to be reduced to 0.5 mg/L or less. Thus, 80 ppm of SBS was dosed for a long-term field test. The required SBS to remove saturated DO 8 mg/L and 0.5 mg/L of residual chlorine in seawater is 53 mg/L. It was reported that SBS injection reduces the pH of seawater to 7.0 or less, which resulted in eliminating sulfuric acid injection. Later, to save a chemical cost, an application of a vacuum deaeration system was proposed. Thus SBS dosing amount was reduced to 20 ppm by installing a vacuum deaeration tower [196].

Due to a lack of information on the SBS-oxygen reaction in seawater, Matsuka et al. [197] investigated factors affecting seawater reaction rate. They evaluated the following factors: salinity, pH, temperature, copper ion, and ethylenediaminetetraacetic acid (EDTA). It was found that the reaction in seawater is much faster than that in pure water. For example, in seawater containing 3.5% salinity at 26 °C, 6 ppm of DO was decreased to almost 0 ppm in about 3 min by dosing 55 ppm SBS. The pH also has a strong effect on the reaction rate. The reaction rate at pH = 6.5 is the highest and about four times as high as that at pH = 5.0. In addition, the positive catalytic effect of copper ions and the negative catalytic effect of EDTA in seawater were observed.

Although the PEC-1000 had high salt and low molecular weight organics rejections, the PEC-1000 was replaced with the TFC polyamide RO membranes. As the performance of the TFC polyamide membrane is not affected by DO, deoxygenation is not necessary [198]. Thus, the oxygen removal in RO application might be of historical interest. However, the information on the oxygen scavenging with SBS is still valuable when considering the cause of membrane deterioration in an SBS/O_2_ system mentioned in Section 8.

The next application of deoxygenation is for anaerobic groundwater. In groundwater, sometimes high levels of iron and manganese ions are contained. Once these ions are contacted with oxygen, colloidal iron- and manganese hydroxides/oxides are generated, which are a cause of RO fouling. Thus, iron and manganese have to be removed in a pretreatment process. The aeration/chemical oxidation followed by filtration has been widely applied [34]. Manganese greensand filtration is one method of providing both iron removal and filtration. However, some reports have shown that direct anaerobic filtration could eliminate iron and manganese fouling [199,200,201]. For example, Beyer et al. [202] reported long-term performance (six and ten years) and fouling behavior of four full-scale NF plants in the Netherlands that treat anoxic groundwater. In addition, it was reported that standard acid-base cleanings (once per year or less) were sufficient to maintain satisfying operation during direct NF of the iron-rich (≤8.4 mg/L) anoxic groundwaters.

For a successful operation with direct filtration, air intrusion should be prevented. However, completely avoiding the air intrusion to RO/NF systems might be difficult. Therefore, lowering the feed pH might be one of the measures to cope with such an event. Hart and Messner [203] mention the reasons that low pH operation eliminates the need to dose a threshold scale inhibitor, slows the rate of iron oxidation, and increases the solubility of any iron oxidation products that did form. Castle and Harn [200] explain that at pH values greater than 5.5, the rate of oxygenation of Fe^2+^ increases by 100 times per pH unit. Therefore, to minimize the chance of iron oxidation, sulfuric acid is added upstream of the membrane system to lower the feed water pH from 7.2 to 5.5 pH units in the Pinewoods Water Treatment Plant in Lee County, FL, USA.

Another option to deal with oxygen intrusion is to add a small amount of SBS. Such a case has been first reported in the Coalinga desalination plant. It is well known that the world’s first large-scale desalination plant using RO was built in California in 1965 at the Coalinga desalination plant [204,205,206]. After startup, several steep declines in production rate occurred due to deposition of ferric hydroxide. It was observed that the feed water contained about 50% saturated DO [205]. Therefore, an oxygen scavenger was added to the feed water to remove the DO and maintain the dissolved iron in the more soluble ferrous oxidation state. Dosing with catalyzed SBS for this purpose was initiated and was found successful in reducing the fouling rate. However, in July 1966, the addition of SBS was stopped from a concern of the chemical cost increase. It should be noted that, currently, using catalyzed SBS is not recommended due to a membrane degradation problem.

Yallaly et al. [207] reported the actual BWRO plant design where the feed groundwater has a TDS concentration of 1150 to 2200 mg/L and has high concentrations of dissolved iron and manganese. Therefore, in addition to lowering feed pH by sulfuric acid, they decided to add SBS in front of RO to treat such feed water directly.

Other cases have been observed in an ion exchange (IEX) softening process using RO pretreatment. Dissolved ferrous iron contained in anaerobic groundwater is effectively removed by standard softening resin [208]. Martin and Kartinen [209] reported the pilot test data to treat groundwater characterized by high TDS content, hardness, and iron and manganese. The combination ion exchange softening and RO process was selected for the pilot test. Although the system was operated in a fashion to exclude air as much as possible, the resin capacity loss was observed and attributed to iron fouling. As well water was pumped directly into the IEX column with no air contact, thus it was speculated that the source of oxygen was the regenerant. To prevent oxidation by the regenerant, SBS was added to the regenerant just before regeneration. This method is expected to benefit from converting the adsorbed ferric iron to more soluble ferrous iron. After taking this measure, the pilot system was operated for 36 regeneration cycles using this regeneration scheme. Over this period of operation, no loss of resin capacity was observed.

The following example is a case study producing process water (160 m^3^/h capacity) in a Ukrainian brewery [210]. The IEX-RO hybrid design was selected to cope with a high level of iron and manganese contained in a groundwater feed. A weak acid cation resin (WAC as H-form) was selected as a softening resin. After taking some measures to avoid air contact, 3–5 ppm of SBS was dosed in the feed water storage tank. As a result, it was reported that the frequency of chemical cleaning of RO elements was significantly decreased to once per 6 months compared with the old system (twice a month), relying on the convention oxidational and filtration process.

Finally, in this section, when applying SBS as an oxygen scavenger in RO processes, there are some points to keep in mind. Bornak [211] raised an issue that sulfite chemistry would probably interfere with other operations at the plant and could pose discharge problems. Overdosing SBS for dechlorination and deoxygenation may cause reduced oxygen levels in the RO brine. Thus, when discharging the RO brine and membrane storage solutions to environmentally sensitive areas, care must be taken [212,213,214].

## 6. Shock Treatment and Sanitization

SBS also acts as an antimicrobial agent [17]. In particular, SBS shows good efficacy for inhibiting the growth of aerobic bacteria as it removes oxygen from the water [36]. Based on this biostatic characteristic, SBS has been widely utilized as an RO shipping solution and preservative, as mentioned in Section 4. Other applications are for shock treatment and RO unit sanitization to prevent or delay biofouling.

### 6.1. Shock Treatment

Shock treatment is the intermittent addition of a biocide into the feed stream during regular plant operation for a limited period [115]. SBS is the most commonly used biocide for this purpose. Historically, research conducted by DuPont has demonstrated that the shock treatment with SBS (500 mg/L) for 30 min twice per day was effective for the hollow fiber polyamide RO [215]. However, the efficacy of SBS as a biocide for seawater is dependent on the use concentration, the exposure time, and the type of micro-organisms present [22,216]. For example, with an exposure time of 30 min at a concentration of 500 ppm, a 99% kill rate was reported for seawater microflora. However, in another case with a high TDS (13,000 mg/L) brackish water, the percent kill was much lower than that obtained with seawater—a 17% kill after 30 min contact at 500 ppm of SBS. Based on these observations, it was suggested that the optimum dosage and exposure time must be determined for each site. To improve the efficacy of the SBS shock treatment, Matani and Kimura [217] invented a new shock treatment method with SBS under an acidic condition for low TDS waters. However, as SO_2_ is generated at low pH, rigorous pH control and monitoring SO_2_ in permeate may be necessary.

Apart from the online shock treatment, offline short-time product water-flushing was also proposed to enhance the plant’s availability [43]. For the hollow fiber polyamide RO, in addition to utilizing standard product water, flushing with 5000 mg/L SBS was suggested to suppress biofouling. This SBS flushing was applied to the SWRO plant in Jabel Dhana, United Arab Emirates [218].

For anaerobic feed water treatment, the SBS concentration was suggested to be increased from 500 up to 2000 mg/L and the shock treatment time from 30 min to 1 h each time, due to less efficiency of SBS in an anaerobic compared to an aerobic environment [219]. Furthermore, anaerobic and sulfate-reducing bacteria are more resistant to SBS than aerobic bacteria. It was also suggested that RO plants are designed to avoid dead ends and stagnant areas where anaerobic bacteria can thrive. If anaerobic bacteria become a problem, offline disinfection and cleaning can kill and remove them from the RO system [22]. For this purpose, quaternary ammonium salts, such as benzalkonium chloride, are reported to be effective in disinfecting anaerobic bacteria for the PEC-1000 membrane [220]. However, as quaternary germicides cause flux losses for the TFC RO membranes, quaternary ammonium salts are not recommended for use as sanitization agents [168].

Next, actual application cases of the SBS shock treatment are introduced. The first case is the 800 m^3^/d capacity demonstration SWRO plant built at Chigasaki, Japan, in 1979 [26,194,221,222]. Two types of RO modules were evaluated (PEC-1000, spiral-wound modules, and CTA hollow fiber modules). As the PEC-1000 has less tolerance of chlorine and oxygen, 80 mg/L of SBS was added to the feed seawater at the beginning. During operation, it was found that differential pressure (DP) increase of the CF was much faster than during the operation of the conventional CA spiral-wound modules. As it was thought that bacteria growth within the CF under a no chlorine condition was a cause of rapid DP increase, the SBS shock treatment was evaluated and implemented (30 min, 500 mg/L SBS, once a day). The DP increase of CF was considerably suppressed by taking this action, and the frequency of CF exchange was extended by about two times.

Later, to minimize SBS consumption for deoxygenation, a vacuum degasifier was installed. Along with this modification, the SBS dosing point was moved after the CF. However, a 500 ppm SBS shock treatment method was continued for these changes. As a result, it was reported that the problem of microbial regrowth was resolved, and the DP increase of the reverse osmosis module was suppressed [26]. As for the PEC-1000 RO, Heyden [223] reported that 500 ppm of SBS shock treatment (twice daily) was applied to a 600,000 gallon per day (gpd) SWRO plant at Tanajib, Arabian Gulf Coast.

The RO plant at Ras Abu Jarjur, State of Bahrain, with a capacity of 46,000 m^3^/d (10 MIGD), was built and started in October 1984 [219,224,225]. As hydrogen sulfide is expected to be contained in the feed groundwater and colloidal sulfur might be generated if it is allowed to contact air, the plant was planned to be operated as a closed system [225]. The shock treatment with SBS was also designed on a dose rate of 500 ppm for half an hour each day. For two years of operation, it was found that the planned SBS shock treatment is less efficient in an anaerobic environment compared to an aerobic environment for bacteria control. Instead, shock dosing with 1000 ppm SBS every second day was established as an optimum bacteria control procedure [224]. Further, periodic SBS soaking of RO membranes with an interval of 6–8 weeks was implemented.

The Boujdour RO plant in Morocco with an 800 m^3^/d production capacity was built to treat beach well water [226]. As a dechlorinating and bacteriostatic agent, 25 ppm SBS was injected continuously to upstream sand filters. Later on, shock-injecting 600 ppm of SBS was applied for 30 min. It was found that applying this shock injection made it possible to attain the same performance as that which has been obtained while using a 25 mg/L continuous injection.

As mentioned above, while there is information that SBS shock treatment well suppresses biofouling, there are also data that its efficacy is questionable or less successful in sustaining bio growth for a long-term operation. For example, it was reported that a major problem at the Al-Birk SWRO plant is biological fouling, although the feed was disinfected with 5.2 ppm chlorine and 30 min of shock treatment every 48 h using 500 ppm SBS [59].

An SWRO desalination plant of 40,000m^3^/d capacity was completed in 1997 at Chatan-cho in the Okinawa Main Island. From the startup, preventing or sustaining biofouling was one of the significant concerns in operation and maintenance. Thus, the plant had decided to apply a shock dosing with 500 ppm SBS (30 min, once per day) into the RO feed water since the startup of the RO plant operation [28]. However, the efficacy of sterilization had been gradually decreasing though it was able to suppress the DP increase of the RO module effectively at the beginning of RO operation. It became clear that the addition of SBS stimulates the growth of microorganisms, including sulfur-oxidizing bacteria, and that the SBS shock dosing cannot stop biofouling [227]. Therefore, they changed SBS shock dosing in July 1999 with sulfuric acid and started with the low pH of 2.5 to 3.0 shock treatment for 30 min. The sulfuric acid shock treatment was effective at the beginning. However, the sulfuric acid shock dosing also tends to decrease its effect after a long-term operation. Furuichi et al. [28] mentioned a possibility that combining two different disinfectants makes it more effective compared with only using a single disinfectant.

Kimura et al. [228] investigated a new membrane sterilization method and compared it with the conventional shock treatment with 500 ppm SBS. They found that most marine bacteria were still alive after contacting 500 ppm of SBS for 2 h. This result does not match the previous results observed in seawater [22]. Next, they tested the new disinfectant and compared it with an actual plant’s SBS shock treatment (1 h per day). It was observed that 0.05 MPa increased the DP within two weeks in the SBS shock treatment. When serious biofouling occurred at the monitoring plant, the consumption of SBS at the RO portion increased with the plant operation time, and the residual SBS in the brine reached almost none, even when the SBS concentration was raised. The authors speculated that particular sulfur-oxidizing bacteria (SOB) can utilize bisulfite ion, or possibly sulfite ion, as a sole energy source. Moreover, several bacteria in the brine were grown in a defined inorganic medium for marine SOB by intermittently adding SBS. The authors conclude that SBS shock treatment is not effective for seawater under these circumstances.

It appears that the efficacy of SBS shock treatment depends on feed water (TDS, AOC, DO, and temperature, etc.) types and pretreatment conditions, etc., and tends to be diminished for long-term operation. Therefore, new disinfectants that enable stable shock treatment have been investigated [228,229,230,231]. First, however, it will be necessary to evaluate its effects, impact on the environment when discharging, and whether it can be used for drinking water during online shock treatment.

### 6.2. Disinfection and Sanitization

Although SBS is used as a biostatic agent to prevent bacterial growth, it is not common to use SBS as a disinfectant. However, SBS has been used as a disinfectant or sanitization agent under some conditions. For example, Redondo and Lomax [41] mention that SBS concentration in the range up to 50 ppm in the feed stream of seawater RO plants has proven effective in controlling biological fouling. In addition, colloidal fouling has also been reduced by this method. However, it was also noted that this method is limited to low- to medium-fouling potential seawater.

RO system sanitization is necessary after CIP in some food and beverage, dairy, pharmaceutical, and microelectronics industries. For this purpose, hydrogen peroxide or a mixture of hydrogen peroxide and peracetic acid has been used [68,232]. For pharmaceutical and kidney dialysis water production, hot water sanitization is also practiced. In specific cases, SBS has been used as a sanitization agent. For example, McDonough and Hargrove [233] evaluated three sanitizing agents (Diethylpyrocarbonate, iodophor, and SMBS) for RO/UF equipment used to concentrate and fractionate cheese whey. They recommended SMBS for overnight shutdown when rapid sterilization is not required. However, when SBS is used for system sanitization, removing heavy metals by acid cleaning should be essential to prevent membrane degradation similar to hydrogen peroxide [161].

As mentioned, the Ras Abu Jarjur RO Plant was operated with SBS shock treatment. However, biofouling gradually built up after two years of operation and affected performance [234,235]. It was found that the primary cause was microorganisms that grew in a storage tank of sodium hexametaphosphate (SHMP). Due to the presence of hydrogen sulfide, SBS was used for sanitization instead of chlorine. The optimum concentration of SBS added to the SHMP tanks was found to be 0.25%, and this concentration was found to control bacteria and does not affect SHMP reversion to orthophosphate. After adding SBS into the SHMP storage tank, the normalized flow rate was sustained longer. For this kind of day tank maintenance, Byrne [236] mentions that bio growth can be prevented by dropping the pH to 4.0 by adding an acid or adding a minimum of 200 ppm of SBS.

## 7. Other Applications: Cleaning and pH Control

Utilizing its reducing action, SBS is also being applied for use in membrane cleaning. This section introduces examples of sulfites application to membrane cleaning and pH adjusters.

When dissolved hydrogen sulfide gas (H_2_S) contained in groundwater contacts with chlorine or DO, H_2_S is oxidized to elemental sulfur or sulfate. Metal sulfides are also formed and can be precipitated on the RO membrane surface. It is said that colloidal sulfur may be challenging to remove, but a solution of sodium hydroxide (NaOH) with a chelating agent, such as EDTA, is an appropriate cleaner [68]. Byrne [21] mentioned that if the foulant is not too heavily composed of elemental sulfur, an acidic solution might be capable of dissolving out the sulfide components. It is also disclosed that a mixed aqueous solution containing sodium sulfite and a wetting agent effectively removes sulfur scales from the metal surface [237]. As shown in Equation (17), sodium sulfite reacts with elemental sulfur (S) and forms sodium thiosulfate (Na_2_S_2_O_3_) in an alkaline solution. Then, the elemental sulfur is dissolved and removed.
Na_2_SO_3_ + S → Na_2_S_2_O_3_(17)

Smith and Whipple [238] reported cleaning test results with SBS. When building a new RO demineralization plant in a paper mill for boiler make-up water, a pilot test was conducted by feeding groundwater in 1987. After about two months of operation, the system pressure drop increased dramatically following system maintenance and possible air intrusion. It was speculated that any air entering the system resulted in sulfur fouling of the membranes due to H_2_S presence in the feed water. Thus, cleaning with a 3% solution of SMBS, which was pH adjusted to pH 8.2 with caustic, was conducted. It was reported that this CIP method was successful in reducing a pressure drop to the normal level. Later, two elements were returned to the manufacturer for cleaning optimization and clement analysis. Then, it was found that copper sulfide was a primary inorganic foulant on the membrane.

Reiss et al. [239] evaluated pretreatment methods containing hydrogen sulfide and elemental sulfur in groundwater. One method was chemical resolubilization with SMBS prior to NF. It was found that 50 ppm of SMBS could reduce turbidity from 40 NTU to as low as 3 NTU, representing over 90% reduction in turbidity. Thus, SBS could be used as a cleaning agent for sulfur and metal sulfides removal based on these data. However, it should be noted that optimum SBS concentration, pH, temperature, and the effect of heavy metals, etc., needs to be identified.

For iron fouling, sodium hydrosulfite (Na_2_S_2_O_4_) is sometimes recommended as a preferred cleaning solution [68,240]. Byrne [21] mentions that the best chance of cleaning the iron is to reduce it from the ferric to the ferrous state, using a strong reducing agent, such as SBS. Once reduced, the iron will readily go back into the solution at lower pH. The optimum pH is about 3.5 to 4.0. Shimizu [241] disclosed an MF membrane-cleaning method for iron oxide and manganese dioxide with SBS.
MnO_2_ + 2NaHSO_3_ → Mn^+^ + 2HSO_4_^−^ + 2 Na^+^(18)

The MF membrane was first cleaned with 2 wt% hydrochloric acid solution, but a good cleaning effect was not obtained. Therefore, the CIP chemical was switched to a 2 wt% SBS solution. After applying the SBS solution for cleaning, the permeate flow rate was reported to be recovered significantly.

Once before, the RO membrane chemical cleaning, which involved SBS and detergent, was used to clean the polyamide hollow fiber membranes [218]. The SBS cleaning with high pH was also reported in RO plants in the Netherlands [242]. SBS was used during the high pH cleaning to achieve anoxic conditions and improve microbial inactivation. As mentioned, biofouling was an issue in the Okinawa SWRO plant. The sulfuric acid shock treatment was effective in decreasing DP for a while but then became ineffective. Therefore, the plant shifted its focus on improving the CIP method. Yamashiro and Goto [243] mentioned a new cleaning procedure. Fouled RO membranes were first soaked in an SBS solution for a fixed time and then cleaned with an alkaline solution after rinsing out the SBS solution. As a result, the DP was decreased drastically, and, consequently, long plant operation became possible after establishing the efficient CIP method.

Regarding the effect of SBS, Yamasato [244] speculated, as follows. From past analysis results, components, such as iron and calcium, exist together in a biofilm on the membrane surface. Thus, for example, when calcium acts as an inhibitory factor for alkaline cleaning, it is considered that the SBS solution with a pH of 3.0 to 4.0 works to remove it and enhances the effect of alkaline cleaning.

Ebrahim [245] reported that biomass and sulfur material sometimes are fouled together on RO membranes in biofouling cases. In this situation, SBS may remove sulfur compounds, such as organic and elemental sulfur and metal sulfides, before alkaline cleaning.

At the end of this section, an SBS role as a pH adjuster is introduced. As mentioned, in some cases, SBS concentration in the range of up to 50 ppm is injected to control biofouling. As a side benefit, no acid is required for calcium carbonate control because of the acidic reaction of bisulfite [41]. Olabarria [20] mentions that in some countries, where it is impossible to have access to acid, it is usual to dose SMBS in well waters. This SBS dosing is used as a disinfectant and reduces the pH.

## 8. Adverse Effects of Sulfites on RO Membranes

Although SBS has been widely used in RO processes, it is reported that SBS has some adverse effects on RO membranes and processes. The following two have been reported to be significant issues and are explained in this section.


RO membrane oxidation;Trigger of biofouling.


### 8.1. RO Membrane Degradation/Oxidation by Reducing Agents

It may sound strange to hear that reducing agents oxidize RO membranes. However, some literature has sporadically reported that some reducing agents deteriorate the RO membrane under specific conditions. For example, the unexamined patent publication (JP2004025027A) [246] disclosed that feed water containing hydrazine (N_2_H_2_) degrades the RO membrane. Hydrazine is a strong reducing agent. A required amount of hydrazine may be added to a cooling water system, such as a circulating cooling water system to prevent slime formation in the water system. When treating the blow-down water from the circulating cooling water system containing hydrazine, the desalination performance of the RO membrane is declined under the specific conditions in which feed water is acidic and contains heavy metals. The ESR-spin trapping experiments demonstrated that the hydroxyl radical is generated during the Mn(III)-catalyzed autoxidation of hydrazine [247]. Similar phenomena have been reported in the RO-SBS systems, especially seawater desalination, where heavy metals play important roles in membrane degradation.

### 8.2. RO Membrane Degradation/Oxidation by Sulfites

Unusual membrane degradation by SBS has been observed without precise cause analysis until Nagai et al. [248] first reported the effect of SBS. The reverse osmosis technical manual (PB80-186950) [31] said that cobalt catalyzed sodium sulfite for dechlorination results in polyamide membrane degradation. The unexamined patent publication (JPS5621604A) [249] mentions the pretreatment method of TFC membranes, consisting of cross-linked furfuryl alcohol. This membrane has no chlorine and oxygen resistivity. Thus, the dissolved oxygen has to be removed by sulfite salts. However, without adding a chelating agent to feed water containing SBS, degradation of TFC membranes occurs, although the deoxygenation rate is expected to be significantly reduced with the chelating agent.

To manage algae growth in the Umm Lujj 2 Desalination Plant, an operational trial with 0.5 ppm chlorination and dechlorination by 5 ppm of SBS was examined. This plant encountered severe membrane degradation even though enough SBS was dosed to eliminate the residual chlorine [47,48,250]. They attributed this result to undetectable halogen compounds generated by the chlorination/dechlorination process and preferential reaction with oxygen due to metal in seawater serving as a catalyst.

Nagai et al. [248] investigated SBS and heavy metals’ effect on colorimetry of residual chlorine (Orthotolidine, DPD) and ORP behavior. Without heavy metals, residual chlorine is completely removed by SBS. However, color was developed by the coexistence of SBS and heavy metals, such as copper (Cu) in the seawater, even without preinjecting chlorine. It was confirmed that Mn^2+^, Zn^2+^, and Pb^2+^ have no effect on the colorimetry at the level of 100 μg/L. Under a similar condition of colorimetry, the ORP of the seawater samples was also measured. Without heavy metals, no sign of ORP increase was observed with 1.35 mg/L of SBS. However, for seawater containing more than 10 μg/L of Cu, the ORP reached over 0.85V. Therefore, they concluded that SBS generates some oxidizing agents in the coexistence of heavy metals, such as Cu and Co, chloride ion, and dissolved oxygen in the solution, in which pH is neutral.

After this report [248], several research works were conducted by mainly Japanese researchers. The seawater desalination facility of Tonaki Village in Okinawa initially used the Polyether Composite RO (PEC-1000) that needed an excessive SBS amount. The SBS injection amount was reduced to a level sufficient for dechlorination when changing the RO elements from the polyether type to the TFC polyamide RO. At this stage, copper sulfate (0.1 ppm as Cu) was continued to be dosed to prevent biofouling. In this situation, membrane performance deterioration was encountered [251]. Similar membrane degradation was also reported in two other pilot tests in Okinawa. In one case, soon after the startup, due to the interaction between copper ions eluted from pump parts and SBS, the ORP was increased. This ORP increase resulted in sharp membrane performance deterioration [252]. Kojima et al. [253] also reported that 30–50 µg/L of copper elution from pump parts induced an abrupt permeate conductivity increase. After controlling feed copper concentration less than 5 µg/L, the permeate quality was stabilized. Talavera [254] reported the destruction of RO membranes due to the addition of a reducer (sodium bisulfite). A 15 ppm of SBS addition caused the degradation of the last elements in pressure vessels. Chlorine was found in the brine flow, but neither in feed nor product. In this case, copper was leached from the garnet in the newly changed sand filters. Talavera said this phenomenon occurred in some plants, and no explanation was made until then. Further in the Tampa Bay desalination plant, unusually high ORP values were detected, even with no free chlorine concentration when overdosing SBS (20 ppm) to the feed [79].

As mentioned above, abnormal oxidants generation was often observed in a dechlorination process by SBS. Similar phenomena were also observed for the ClO_2_/SBS system. In some seawater desalination plants, intake ClO_2_ dosing has been applied to reduce DBPs, such as THMs in the product water. However, an undesirable ClO_2_^−^ ion is generated by dosing ClO_2_ in a seawater matrix [79,114,155]. Under this situation, oxidant generation or ORP increase were observed when dosing SBS to eliminate ClO_2_ and ClO_2_^−^ in RO permeate and brine [79,114]. Tanaka et al. [114] found that SBS generated some oxidizing agents in the case of the coexistence of ClO_2_ in seawater, such as RO feed water and RO brine water. At the neutral pH in the range of 6.7–7.2, 10 mg/L of SBS did not reduce chlorite ion to chloride, and SBS generated some oxidizing agents with a 5 min contact period. They also found that sodium thiosulfate reduced chlorite ion to chloride at neutral pH (6.7–7.2) in RO brine water without forming oxidizing agents. The same conclusion was drawn by Doñaque et al. [157] that SBS does not easily remove chlorite ion in the presence of metal ions, such as copper and Cu(II), and the concentration of chlorite ion is increased from the high concentration of chloride ion in seawater.

This unusual membrane degradation has been mainly reported for seawater desalination and wastewater reclamation under neutral pH conditions. However, Nada et al. [15,255] reported the case of the second-pass RO at high pH. Shuqaiq Phase II—Independent Water and Power Project (IWPP) is located on the southwestern coast of Saudi Arabia. The seawater reverse osmosis (SWRO) plant can produce 212,000 m^3^/d of desalinated drinking water. The process of the Shuqaiq plant is based on the two-pass process, where the feed pH of the second-pass BWRO is adjusted to pH = 10.0 to produce a boron concentration water less than 0.5 mg/L. The first-pass SWRO used cellulose triacetate hollow fiber modules and applied the intermittent chlorine injection (ICI) method to eliminate biofouling of the membranes. Chlorine was injected for 3 h per day by the operation without SBS injection to filtered water. Under this condition, 20 mg/L of SBS was continuously dosed in front of the BWRO to eliminate residual chlorine. Under these operational conditions, the BWRO membranes experienced severe performance degradation during the commissioning stage, as shown in Figure 5.

The permeate conductivity showed a sharp increase for one week, and the second-pass BWRO membranes completely lost salt rejection. At the same time, online ORP values that are measured at the upstream of the BWRO membrane by two ORP meters also increased gradually. After observing the membrane degradation, an autopsy analysis of a deteriorated membrane was conducted. The electron spectroscopy for chemical analysis (ESCA) was performed for the damaged membrane. No Cl and Br halogen atoms were detected. Thus, the idea of membrane damage by halogen oxidation was eliminated.

Up to this point, several cases were examined in which membrane degradation was observed during operation. However, in some specific circumstances, the performance was declined during SBS preservation. It was reported that the SWRO membrane rejection was decreased from 99.64% to 99.22% during 10 days of storage with 1000 ppm of SBS, where the SWRO elements were operated with feed water containing 10 ppb copper, 10 ppb chromium, and 10 ppb of nickel for 24 h [184,256]. Although exact causes were not identified, a similar failure was observed for the aromatic polyamide hollow fiber RO module [185]. A permeate conductivity increase was observed in some BWRO membranes preserved in SBS solution for about 23 days. The failure was confirmed by the intrinsic viscosity data, where the damaged membrane was found to have lower intrinsic viscosity compared with the good performance membrane. When analyzing the membrane surface by SEM-EDX, Fe and Cr were detected.

As mentioned, feed ORP increase was observed associated with SBS-originated membrane oxidation. An unexamined patent publication (JPH09290259A) [257] reported feed and concentrate ORP values with different feed SBS concentrations. The concentrate ORP prominently increases compared with RO feed water when the dosing amount of SBS is in the 3–11 ppm range. In these concentrate solutions, residual chlorine (o-Tolidine method) is detected. Traditionally, only feed ORP has been monitored to prevent membrane oxidation by residual chlorine. However, for the membrane degradation by SBS, it is essential to monitor both feed and concentrate ORP. Hu and Maeda [258] observed the ORP increase in a wastewater reclamation plant of a purified terephthalic acid manufacturer in Taiwan, where cobalt was contained in feed water to RO. It was reported that overdosing of SBS increased concentrate ORP and created an oxidation atmosphere within the RO stages.

### 8.3. Mechanism Membrane Degradation with SBS

While there are many reports of phenomenological membrane degradation by SBS, there are few reports on the exact mechanism. An unexamined patent publication (JPH07-328391) [184] mentions a mechanism of membrane deterioration. When heavy metals, such as Cu, Co, Cr, and Ni, are contained in the feed water, a bisulfite ion is converted to a sulfite radical. A strong oxidant of persulfate is generated from a sulfite radical. Further, this persulfate reacts with chloride ions in the feed solution to generate perchlorate ions and chlorine. Iwahori et al. [259] suggests the reaction of [SBS-Cu^+^] + Cl^−^ → [SBS-Cu] + Cl∙ (chlorine-free radical) in seawater coexisting with copper.

In an effort to solve the membrane oxidation problem and elucidate the mechanism of membrane degradation by SBS, Nada et al. [15] first conducted literature searches [260,261,262]. They observed many similarities for DNA damage by sulfite [263,264,265,266,267,268,269,270,271,272,273,274,275,276,277,278]. For example, Kawanishi et al. [268] reported the reactivities of sulfite (SO_3_^2−^) with DNA in the presence of metal ions and attributed the site-specific DNA damage to ∙SO_4_^−^ radical generated from sulfite autoxidized in the presence of Co^2+^. Recently, the oxidative degradation of organic compounds conjugated with sulfite oxidation has been investigated [279,280,281,282,283,284] and is proposed as a new type of advanced oxidation process (AOP) [285,286,287,288,289,290,291,292,293,294,295,296,297,298,299]. In this context, autoxidation of sulfite catalyzed by heavy metals is postulated. Most of the published reaction mechanisms for the homogeneous transition metal-catalyzed autoxidation of S(IV) oxides suggest radical mechanisms that are based on the scheme given by Backström [260,300]. Radical scavengers, such as mannitol, tert-butyl alcohol, ethanol, and hydroquinone inhibit, the overall S(IV) oxidation process (negative catalysis) [260]. A simplified reaction scheme of autoxidation of sulfite/bisulfite is shown in Figure 6 [260,261,286,289,291,298,301,302,303].

The reaction of transition metal ions (M^n+^) with HSO_3_^−^/SO_3_^2−^ is an initiation step to form sulfite radical (∙SO_3_^−^). In the propagation chain reaction, three essential radicals are formed, that is, sulfite radical (∙SO_3_^−^), peroxymonosulfate radical (∙SO_5_^−^), and sulfate radical (∙SO_4_^−^). Shi et al. [304] indicated that hydroxyl (∙OH) radical is also generated in the sulfite oxidation pathway. Liang et al. [284] suggested the following reactions produce the hydroxyl radical (∙OH) from ∙SO_4_^−^.
All pHs: ∙SO_4_^−^ + H2O → SO_4_^2−^ + ∙OH + H^+^(19)
Alkaline pH: ∙SO_4_^−^ + OH^−^ → SO_4_^2−^ + ∙OH(20)

Radical scavenging tests used to identify predominant radical species suggested that the sulfate radical (∙SO_4_^−^) predominates under acidic conditions, and the hydroxyl radical (∙OH) predominates under basic conditions. Liang and Su [305] reported the following oxidant presence in various pH:
∙SO_4_^−^ is the predominant radical at pH < 7.0; Both ∙SO_4_^−^ and ∙OH are present at pH 9.0; ∙OH is the predominant radical at a more basic pH (i.e., pH 12.0).

Further, although not shown in Figure 7, side reactions producing persulfates (peroxomonosulfate and peroxydisulfate) from ∙SO_5_^−^ and ∙SO_4_^−^ radicals have also been reported [260,286,303].
∙SO_5_^−^ + HSO_3_^−^/SO_3_^2−^ → ∙SO_3_^−^ + HSO_5_^−^/SO_5_^2−^(21)
∙SO_5_^−^ + ∙SO_5_^−^ → S_2_O_8_^2−^ + O_2_(22)
∙SO_4_^−^ + ∙SO_4_^−^ → S_2_O_8_^2−^(23)

Under high chloride ion conditions, such as seawater, it is reported that the hypochlorite ion (OCl^−^) is also generated by reacting persulfate (peroxomonosulfate ion) a with chloride ion [157,184,306].
SO_5_^2−^ + Cl^−^ → SO_4_^2−^ + OCl^−^(24)

Table 9 shows the oxidation potential of relevant chemical species during SBS autooxidation. Hydroxyl radical and sulfate radical are two of the strongest oxidants available with oxidation potentials of 2.8 V and 2.6 V, respectively.

Thus, it was thought reasonable that the RO membrane degradation by SBS is related to sulfite autoxidation in the presence of heavy metals. Assuming the reactions occur in a heavy metal-fouled membrane surface, the following schematic oxidation process was drawn, as shown in Figure 7 [255].

### 8.4. Factors of Membrane Degradation with SBS

Regarding deoxygenation with SBS, Matsuka et al. [197] elucidated the effect of salinity, bicarbonate ion concentration, pH, temperature, cupric ion, and EDTA. Osta and Bakheet [47] pointed out that bisulfite reaction with oxygen at the Umm Lujj SWRO plant is fast, due to the following factors: (a) metal in seawater serving as catalysts; (b) high ionic strength; (c) specific pH; (d) high bicarbonate concentration; and (e) high temperature. Nagai et al. [248] indicated several factors concerned with the generation of an oxidizing agent. Kawada et al. [252,309] revealed the importance of NaCl concentration, bicarbonate ion, and pH. The controlling factors affecting membrane degradation by SBS are summarized in Table 10.

#### 8.4.1. Effect of Heavy Metals

Heavy metals have a critical role in the initiation step forming the ∙SO_3_^−^ radical as a catalyst. Brandt and Eldik [260] summarize the catalytic activity of transition metal ions in the oxidation of S(IV) oxides. They reported that iron and manganese are the most effective catalysts in the oxidation of S(IV) oxides in an aqueous solution. Thus, heavy metals’ effect on the RO membrane degradation was intensively investigated. These findings are summarized in Table 11.

It appears that copper ion is the most harmful ion in seawater desalination. There are many reports that copper ion causes the SBS-originated membrane degradation. In the case of 2.5 ppb of copper ion, the ORP was the same as raw seawater (0.6V). However, for more than 10 ppb of Cu, the ORP reached over 0.85 V [248]. Kawada et al. [309] reported an ORP increase that was not observed at the copper ion concentration of 1 ppb, which is about the copper ion concentration in natural seawater. Instead, they saw the ORP-increase with over 5 ppb of copper ion. The unexamined patent publication JPH0957067A [310] disclosed that when a small amount of copper eluted from RO equipment materials exceeds 2 ppb, an oxidizing substance is generated. Iwahori et al. [259] pointed out that piping, fitting, pump, and instrumental materials consisting of copper metal or alloy are the origin of copper. In addition, they mentioned that pretreatment chemicals, such as FeCl_3_ and SBS, contain impurity ingredients.

Cobalt was first reported as a harmful metal when SBS is applied for dechlorination [31]. The catalytic activity was observed for the chemical plant wastewater containing Co [258].

**Table 11 membranes-12-00170-t011:** Effect of heavy metals on SBS originated membrane degradation.

Heavy Metal		Concentration	pH	Positive or Negative		Reference
Cupper	Cu			(+)		[157,248,251,252,253,309,310]
		<2.5 ppb	6.5	(−)	Colorimetry	[248]
		>5 ppb		(+)	ORP	[309]
		1 ppb		(−)	ORP	[309]
		30–50 ppb		(+)	Membrane	[253]
		0.1 ppm	10	(+)	Membrane	[255]
Cobalt	Co			(+)		[31,258,309,310]
		<2.5 ppb	6.5	(−)	Colorimetry	[248]
Tin	Sn			(+)	ORP	[309]
Iron	Fe	10 ppm	10	(−)	ORP	[255]
		Precipitated	9.5 9.6	(+)	Membrane	[311,312]
		1.5 mg/L	6.7	(+)	Membrane	[313]
Manganese	Mn	100 ppb	6.5	(−)	Colorimetry	[248]
Fe/Mn mix		30/30 ppb	10	(+)	Membrane	[314]
Zinc	Zn	100 ppb	6.5	(−)	Colorimetry	[248]
Lead	Pb	100 ppb	6.5	(−)	Colorimetry	[248]

ORP: Oxidation redox potential measurement. Colorimetry: Residual chlorine measurement (Orthotolidine and DPD).

As for the effect of iron, mixed results were obtained. When adding 10 ppm of Fe^3+^, no ORP increase was observed in the testing solution [255]. However, Ferrer et al. [313] reported that 1.5 ppm of Fe(III) resulted in a severe increase in chloride permeation for RO membranes. Unexamined patent publications (JP2005246282A & JP2008029965A) [311,312] mention that precipitated iron induces membrane degradation at high pH. As the oxidation reaction is presumed to occur near the membrane surface, it is necessary to pay attention, not only to the concentration of heavy metals in the feed water but also to the precipitated heavy metals on the membrane surface.

#### 8.4.2. Effect of SBS Concentration and DO Concentration

DO directly contributes to the first propagation step reacting ∙SO_3_^−^ with oxygen, as shown in Figure 7. As mentioned in Section 5, because SBS is used as an oxygen scavenger, the SBS concentration should significantly impact residual oxygen and membrane degradation. Nagai et al. [248] measured the seawater ORP by changing SBS concentration from 2.7 mg/L to 40 mg/L. The ORP increase did not occur in the case of 40 mg/L SBS. A similar result was observed for a 3.5% NaCl solution containing 146 ppm NaHCO_3_ [309]. When the SBS concentration is relatively low, the ORP increases with the increase of SBS concentration. However, it sharply decreases at the SBS concentration of between 40 and 100 ppm. It was also confirmed that when higher ORP is observed by adding SBS, the residual SBS could not be detected in the sample. As previously mentioned, the prominent delta-ORP increase was observed for 3–10 ppm SBS concentration ranges [257]. However, when the amount of added SBS reaches 40 to 100 ppm, an increase of the delta-ORP was not observed and free chlorine was also not detected. These three results are summarized in Figure 8.

The same result was observed at the higher pH and low salinity conditions [255]. Flat sheet BWRO membranes were first soaked in 1 mg/L of Cu^2+^ solution at pH 10.0 for 4 h in advance. After that, the fouled membranes were set in the flow cell, and the fresh SBS solutions adjusted to pH = 10.0 were continuously supplied for three days. After that, membrane performance was evaluated. The result of the salt passage is shown in Figure 9. It demonstrates that salt passage became higher than normal when SBS concentration was 7.5 mg/L to 50 mg/L [255]. The test results suggested that the membrane deterioration might be inhibited if the SBS dosing was well controlled and adjusted to the design value of 0.75 mg/L to eliminate residual chlorine.

Based on these results, it was clarified that the effect of SBS concentration relates to deoxygenation. By adding enough SBS to remove DO, membrane degradation can be prevented. Theoretically, 8 mg/L of DO can be removed by 52 mg/L of SBS. In seawater containing 3.5% salinity at 26 °C, 6 ppm of DO was decreased to almost 0 ppm in about 3 min by dosing 55 ppm SBS. It is also reported that adding 80 mg/L of SBS completely removes DO in the field test [197]. When applying a vacuum deaerator, 20 mg/L SBS was enough to eliminate residual oxygen. This analysis is consistent with the data shown here, reporting that there exists a threshold level of SBS concentration between 50–70 mg/L ranges in seawater.

#### 8.4.3. Effect of Feed pH and Bicarbonate Concentration

It has been reported that solution pH affects sulfite oxidation [47,197,309]. It was found that the deoxygenation reaction rate by SBS at pH = 6.5 is highest for 3.5% NaCl containing 150 ppm bicarbonate ion and about four times as high as that at pH = 5.0 [309]. A suitable pH value for the reaction is between 6.0 and 8.0 [197]. For the transition metal-catalyzed S(IV) autoxidation process, it is reported that the oxidation rate is influenced by the pH-dependent distribution of the metal ions and of the S(IV) species [260]. HSO_3_^−^ is about 20–40 times less reactive than SO_3_^2−^. As bisulfite is a dominant species at pH less than 6.0, more minor oxidative damages to RO membranes can be expected at acidic conditions. Nagai et al. [248] measured the ORP by varying seawater pH from 7.0 to 9.7. They mentioned that with increasing pH, the ORP decreased, and at pH 9.7, an increase of ORP was not observed. However, this result does not seem to match subsequent research results. Nada et al. [15] reported a steep ORP increase at pH 10.0.

For the test to remove chlorine dioxide and chlorite in seawater, SBS did not reduce chlorite ion to chloride, and SBS generated some oxidizing agents with a 5 min contact period at a neutral pH in the range of 6.7–7.2 [114]. Chlorite ion was reduced to chloride at low pH, and SBS did not generate oxidizing agents at pH below 5.2. Kawada et al. [309] also reported the same phenomena. At a pH below 4.0, the ORP increase was not found even after SBS and copper ion were added. These two results are summarized in Figure 10.

Regarding the ORP data, as the ORP naturally has pH dependence, the delta-ORP (difference before and after SBS addition) was plotted. It seems that no oxidant is generated at pH below 5.0 for high salinity solutions. The two unexamined patent publications, JPH07308671A and JPH07328392A [256,315], claim the lowering of RO feed pH to less than 4.0 and 6.5, respectively.

Bicarbonate ion plays an essential role as a buffering effect to maintain solution pH after dosing SBS. During the deoxygenation test with SBS, two sample solutions were compared. One is an unbuffered 3.5% NaCl solution, and the other is a buffered solution with 150 ppm of HCO_3_^−^. Although pH values of both solutions are equally controlled to be 8.0 before dosing with SBS, each solution showed a different pH value after dosing. The pH value decreased from 6.0 to 5.0 in about 3.4 min after the unbuffered solution’s SBS dosing. On the other hand, the pH value was kept between 6.8 and 6.6 during the reaction for the buffered solution. As a result, 6 ppm of DO was decreased to only about 4 ppm in 3.4 min by dosing 50 ppm of SBS. For the buffered solution, 6 ppm of DO was reduced to almost 0 ppm by dosing with the same amount of SBS. Kawada et al. [309] measured the ORP as a function of NaHCO_3_ concentration. The ORP becomes highest at about 100 ppm of NaHCO_3_, and then it gradually decreases at a higher concentration. To prevent membrane degradation, adding a decarbonation pretreatment process might be better to lower feed pH.

#### 8.4.4. Effect of Salinity and Any Other Ions (Chloride)

The deoxygenation reaction rate is increased by sodium chloride concentration [197]. The reaction rate constant in sodium chloride solutions, under the condition that the pH value of the solution is kept constant by bicarbonate ion, increases in proportion to the concentration of sodium chloride. The salinity of the seawater is found to have an effect on increasing the reaction rate. For the model solution test, no ORP increase was observed for deionized water and 146 ppm of NaHCO_3_ solution at final pH 6.13 and 7.36, respectively [309]. Furthermore, when copper ion and SBS were added to a relatively low concentration of NaCl solution, an ORP increase was not found. The ORP increase took place only when NaCl concentration was higher than about 1000 ppm

However, even under low TDS conditions, the membrane deterioration was observed for wastewater containing Co^2+^ [258], the second-pass permeate at a high pH 10.0 [255] and Fe(III)-SBS system [313]. Thus, the effect of salinity and chloride ion is not presently clear. A detailed analysis considering the effect of concentration polarization may be necessary.

### 8.5. Countermeasures of Membrane Degradation Originated from SBS

As the membrane degradation induced by SBS is considered to be initiated with transition metals, thus the first step is to remove such heavy metals in a pretreatment step and select appropriate construction materials and chemicals [259]. However, it is challenging to completely eliminate the heavy metals from feed water. Therefore, many other preventive measures have been proposed and are summarized in Table 12.

The most common method proposed is to add chelating agents to feed solutions. It is also recommended to add the chelating agents to RO element preservatives [184]. Many types of chelating agents are proposed. However, EDTA is the most common and effective chelating agent associated with SBS oxidation prevention. For deoxygenation, adding 500 ppm (pH 4.0) and 5000 ppm (pH 3.6) EDTA could prevent oxygen removal in 3.5% seawater [197]. Kawada et al. [309] studied the effectiveness of sodium hexametaphosphate (SHMP) and EDTA as metal sequestering agents. Both of the chelating agents exhibited the inhibiting effect of the ORP increase. Similar effects were obtained for the commercial scale inhibitor, Flocon 100 (polyacrylic acid-based). However, its effect seems to be considerably lower than that of EDTA. The order of effectiveness was EDTA > SHMP > Flocon 100. The required chemical concentration to inhibit the ORP increase of the copper and SBS-added test solution (3.5% NaCl with 146 ppm NaHCO_3_) was 20 to 25 times with EDTA, about 100 times with SHMP, and about 200 times with Flocon 100, compared to the copper ion concentration. It was reported that 10 ppm of EDTA was enough to prevent ORP increase for 3.5% NaCl solution containing 400 ppb of Cu^2+^ [315]. By employing flat sheet membranes, the effectiveness of SHMP and EDTA proved, as shown in Table 13 [184], that 5 ppm of EDTA was enough to prevent membrane degradation.

**Table 12 membranes-12-00170-t012:** Countermeasures to prevent SBS-originated RO membrane degradation.

Preventive Countermeasures	Reference
Addition of chelating agents (e.g., EDTA, SHMP)	[249,252,309,315]
A scale inhibitor having a reducing function Phosphorous acid-based or phosphonate compounds	[316]
Addition of chelating agents to SBS preservative	[184]
Addition of radical or oxidant scavengers Add thiosulfates	[306]
Remove oxygen (e.g., vacuum degasification)	[257]
Preventive cleaning with acids to remove heavy metals	
Measure the brine heavy metals—Cu, Co	[310]
Monitor the brine ORP	[257,317]
Operate under lower pH (e.g., <pH 5.2)	[114]
Operate under lower pH (e.g., <pH 4.0, relates to HCO^3+^)	[315]
Operate and preserve under pH < pH 6.5 and/or <30 °C	[256]
Maintain feed or concentrate Cu or Co concentration < 2 µg/L	[310]
Alternative reducing agents for dechlorination	
Reducing organic or phosphorus compounds (e.g., L-ascorbic acid, sodium hypophosphite)	[318]
Sodium thiosulfate	[114]
High pH second-pass RO	
Pretreatments, selected from the following processes: iron or manganese removal, decarbonation, chelating agents, and scale inhibitors	[314,319]
Place <10 µm cartridge filter in front of second-pass RO	[311]
Adjust SBS concentration for the second-pass RO < 0.5 mg/L	[312]
Phosphonate scale inhibitors	[15,255,320]

EDTA was thought to be an effective antioxidant in the membrane systems. However, under higher pH (e.g., pH 10.0) conditions, membrane oxidation cannot be prevented [255]. For the beaker test, it was found that 1 mg/L of EDTA was enough to inhibit the ORP increase. On the contrary, degradation could not be entirely prevented when injecting 1 mg/L of Na4-EDTA into the flow cell. Besides SBS, an appropriate antiscalant needs to be injected at the Shuqaiq desalination plant to suppress the second-pass carbonate scaling. Along with this situation, an antiscalant (Genesys LF) was also tested. The membrane degradation was entirely inhibited by dosing the combination of 1 mg/L Na4-EDTA with 1 mg/L of the antiscalant. Interestingly, only 1 mg/L of antiscalant had the same effect. It was thought that Genesys LF, neutralized phosphonate, has a strong chelating effect at high pH. This result is consistent with the radical generation in alkaline peroxide [321]. In the presence of the chelating agents, such as diethylenetriamine pentamethylphosphonic acid and sodium salt (DTPMP) and various transition metals, the generation of ∙OH radical is much reduced, at pH 10.0. On the other hand, EDTA exerts little protective effect. This phenomenon was attributed to differences in stability constants and speciation effects. After finding the simultaneous effects as a scale inhibitor and a chelating agent, the use of phosphonate compounds is suggested to the SBS system [255,320].

It is said that radical scavengers, such as mannitol, tert-butyl alcoho1, ethano1, succinic acid, and hydroquinone, inhibit the overall S(IV) oxidation process (negative catalysis) [197,260]. This action can be interpreted as evidence for a free radical chain reaction during the transition metal-catalyzed autoxidation of S(IV), as shown in Figure 6. Therefore, it might be natural to consider that the addition of such radical scavengers could prevent RO membrane oxidation. For example, the patent publication JP2020049418A [306] disclosed that the addition of thiosulfate effectively avoids membrane oxidation. It is mentioned that thiosulfate neutralizes strong oxidants generated from a catalytic reaction of SBS or forms a complex with a transition metal. Rochelle et al. [322] mentioned that thiosulfate appears to function as a free radical scavenger rather than a complexer of S(IV) (sulfite plus bisulfite) or metal ions. However, using thiosulfates as a dechlorinating agent might be straightforward and simpler, as mentioned by Tanaka et al. [114].

The next countermeasure is to conduct preventive CIP. This approach first needs to detect a sign of oxidation. For this purpose, two methods are proposed. The methods are to monitor the heavy metal concentration, e.g., Cu and Co, or the ORP in the brine [257,310,317]. It is disclosed that membrane degradation can be prevented by setting 300 mv of ORP as the control value and conducting the timely CIP with citric acid [317]. Changing the operational conditions, such as pH and temperature, might be an alternative approach. As it has been reported that neither oxidant nor ORP increase was observed at lower pH, operating an RO system with lower pH can be considered an option for system design [114,256,315]. SBS has been widely used for dechlorination in RO systems due to its reliability and economic aspects. However, from the point of preventive membrane degradation, alternative reducing agents were proposed. Tanaka et al. [114] found that sodium thiosulfate does not form oxidizing agents. Thus, using sodium thiosulfate might be an option for dechlorination. Other types of reducing agents were also proposed, such as reducing organic or phosphorus compounds (e.g., L-ascorbic acid, sodium hypophosphite) [318].

As mentioned previously, unexpected membrane oxidation was encountered in the Shuqaiq seawater desalination plant, where the second-pass BWRO was operated at pH 10.0 to increase boron rejection. Usually, a dechlorination step is not necessary for the second-pass RO. However, when using the cellulose acetate SWRO in the first pass and applying intermittent chlorination, the dechlorination process becomes necessary for the second-pass BWRO. Originally, heavy metal fouling was not anticipated for the second pass. However, heavy inorganic foulants were detected on the membrane surface [255]. This issue may come from the following two reasons. First of all, residual soluble heavy metal ions at lower pH are precipitated at high pH conditions. Those precipitated or colloidal metals cannot be removed, as many second-pass RO units are not equipped with a cartridge filter [255] in front of the BWRO. Several patents were filed [311,312,314,319,320] to address those specific issues at high pH conditions. The US open patent (US 2004/0050793 A1) [314] indicated the importance of removing bicarbonate ion in the pretreatment steps in addition to Fe and Mn. An unexamined patent publication (JP2005246282A) [311] claims that the feed water is filtered with a cartridge filter having a pore size of ≤10 µm (preferably ≤ 5 µm) to remove heavy metals.

## 9. SBS Acts as a Trigger of Biofouling

Once before, Flemming et al. [323] referred to biofouling as the Achilles heel of membrane processes. After that, there has been a lot of progress in preventing biofouling, e.g., developing low fouling RO membranes/elements, utilizing LP membranes as pretreatment, non-oxidative biocides, monitoring techniques, and so on. However, biofouling continues to be one of the significant obstacles to achieving steady RO operations. Many review articles have reported how to tackle unresolved issues [36,216,324,325,326,327,328,329,330,331,332,333,334].

Biofouling is a very complex membrane phenomenon. Thus, one cannot attribute a single factor to a cause of a biofouling initiator. However, chlorination–dechlorination has been raised as one of the potential causes of enhancing biofouling. As described in Section 3, continuous chlorination–dechlorination has been implemented in RO feed waters as a part of pretreatment, such as open-intake surface seawater. It is reported that after dechlorination with SBS, bacterial activity increased, resulting in an increase in biofouling [335]. Sometimes this phenomenon is referred to as “aftergrowth” [43,57,104,336,337]. Because chlorination does not sterilize the seawater, the surviving bacteria can grow quickly (aftergrowth), resulting in biofouling. Kimura et al. [228] measured in situ viable bacteria counts in an SWRO plant in Japan, where continuous chlorination–dechlorination (SBS) was applied. Viable cells were rarely detected from the seawater samples in the presence of chlorine. However, once adding SBS, bacteria drastically increased, especially before the RO element and its brine. Other biofouling indices, ATP, HPC, and modified biofilm formation rate (mBFR) values were also increased after the SBS addition [338,339]. The following factors have explained the aftergrowth:Surviving bacterial quickly grow under no chlorine conditions and no competition [104,340].Chlorine oxidizes NOM/humic substances, and assimilable organic carbon (AOC) is formed that can be considered as nutrients for surviving bacteria [104].Non-viable microorganisms following chlorination act as a nutrient source [216,341].Produce extracellular polymeric substances (EPS) as a defense mechanism when bacteria are exposed to chlorine [216,339].

Under the hypothesis that chlorine degrades organics in the seawater feed to produce nutrients (AOCs), Moch et al. [43] proposed to operate plants without chlorination. These kinds of new disinfection methods, e.g., intermittent chlorination and chloramination, were proposed and have been commonly implemented [43,342,343,344,345,346,347]. Hamida and Moch [342] reported that by eliminating the continuous addition of chlorine to seawater feed streams, biofouling had been contained, and plant availability has been maintained at over 90% in ten plants located in the Arabian Gulf, the Indian Ocean, and the Caribbean Sea. Furthermore, there have been reports that intermittent chlorination effectively prevents biofouling to brackish RO (BWRO) plants [343]. As a result, continuous chlorination has become less common, and intermittent chlorination–dechlorination appears to be more frequently applied to RO, especially seawater desalination [347].

In the above phenomena for aftergrowth, the roles of SBS have received insufficient attention. However, in some cases, it was observed that high doses of SBS were not providing improvements in the SWRO operation. But instead, the opposite of what was expected was observed [348]. Then, after fundamental analysis, Ito et al. [339] speculated that SBS alone could also trigger biofouling. In terms of such a negative impact of SBS on RO elements, the following factors were postulated:Creating an anaerobic environment to enhance anaerobic bacterial growth, such as sulfate-reducing bacteria (SRB);SBS enhance some types of bacteria as food, such as sulfur-oxidizing bacteria;SBS increase AOC due to organic oxidation.

First of all, it is said that excessive SBS dosing during dechlorination consumes some of DO and creates an anaerobic environment that increases the potential for increased anaerobic biological growth. Byrne [75] stressed an adverse effect of SBS that is responsible for heavy slime formations. A definitive symptom of this is the sulfur dioxide, rotten-egg smell noted when membrane vessels are opened. During the commissioning of the Point Lisas SWRO plant in Trinidad and Tobago, the typical black color and slight stench were detected; the sulfur scent was also observed during CIPs. Hence, it was speculated that the dechlorination step using SBS is a source of biofouling [349]. However, additional factors have to be considered from the following points: DO level after SBS dosing and whether anaerobic feedwater is a friend or foe.

Typically surface seawater contains 5–8 ppm of DO depending on salinity and temperature. To eliminate such a level of DO, at least 32–52 mg/L of additional SBS must be dosed to the feed water. In actual desalination plants, the dosing amount of SBS is much less than those numbers. Thus, the bulk feed water contains some oxygen and is still in aerobic condition. However, it is well known that when a biofilm forms, anaerobic conditions exist at the base of the biofilm [350]. When the SWRO plant in Santa Barbara, Curacao, encountered a biofouling problem and found SRB during the element autopsy, Dorival et al. [62] postulated that facultative bacteria in the anaerobic regions begin metabolizing bisulfite.

Another issue is whether anaerobic feed water is a friend or foe for RO plant operations. There are reports that anaerobic groundwater NF/RO plants have operated well without prominent biofouling [199,200,201,202,234,235,351]. In the case of the Ras Abjajur brackish water treatment plant in Bahrain, after solving initial operational difficulties, e.g., biofouling in an antiscalant tank, the plant operated smoothly [234,235]. The same successful operation was reported in NF plants in the Netherlands. Beyer et al. [202] mentioned that when compared to aerobic NF and RO systems (e.g., aerated groundwaters or surface waters), the operation and performance of the anaerobic installations (with minimal pretreatment) could be described as very stable. This phenomenon is explained by the higher growth rates and yields associated with aerobic growth versus anaerobic growth. Aerobic growth is limited in the absence of oxygen because these microorganisms cannot metabolize biodegradable organic material (BOM) [351]. Relating to SBS overdosing, some insight can be obtained from the following case. As mentioned, as the PEC-1000 is less tolerant to DO, the DO has to be removed. By adding 80 ppm of SBS, the deoxygenation is completed. Under this condition, a pilot with more than 7000 h was reported successful without any anaerobic bacteria fouling [196]. As for the actual SWRO plant operation in the Gulf, the plant encountered biofouling with SRB after two years of operation. However, it was reported that 0.1% benzalkonium chloride was able to remove the hydrogen sulfide smell [220]. When reviewing data, even though SBS overdosing causes biofouling with anaerobic bacterial, e.g., SRB, anaerobic bacteria growth itself might be managed by proper CIP.

The following discussion points out that SBS enhances some types of bacteria growth, such as sulfur-oxidizing bacterial (SOB) and SRB under aerobic or anaerobic environments, respectively. It is known that certain types of anaerobic bacteria obtain energy from the disproportionation of inorganic sulfur, such as thiosulfate or sulfite [352]. Adding SBS to a brackish subsurface water transport pipeline increases the sulfur-disproportionating bacteria (SDB) and Desulfocapsa together with the SRB [353]. As for the RO plants, Ito et al. [339] elucidated the effect of SBS on the biofouling potential of feed seawater by conducting mBFR measurements. During a pilot test, no chemical was added to surface seawater with UF pretreatment in one skid. For another skid, 1 ppm of SBS was continuously injected into the feed seawater. It was demonstrated that a small amount of SBS could stimulate microorganisms and increase the mBFR value to 60 pg-ATP/cm^2^/day, which is about twice higher than that of a no-chemical dosing operation 29 pg-ATP/cm^2^/day. In another approach, assuming that SBS might be a sulfur source for microorganisms, an aprA (adenosine-5-phosphosulfate reductase) gene clone library was constructed, which encodes the key enzyme involved in both sulfate reduction and sulfur oxidation processes [354]. In RO supply water with chlorination-dechlorination, specific aprA sequences were predominant among the diversities found for respective supply water, indicating that the selection for bacteria involving in sulfur metabolism would have been completed.

Furthermore, from an actual SWRO plant and pilot trials, a higher concentration of sulfur (S) was detected from membrane foulants by ICP analysis [338,355]. Thus, the authors suspected that a higher concentration of S in the foulants could be attributed to continuous SBS dosing before the SWRO train. Based on these analyses and observations, it might be reasonable to consider that the overdosing of SBS induces biofouling.

Kimura et al. [228] first reported sulfur-oxidizing bacteria (SOB) as a cause of biofouling in the SWRO plant. In this plant, the excess amount of SBS was added to completely remove the residual free chlorine. The consumption of SBS at an RO portion increased during plant operation, and the concentration of SBS in the brine reached almost zero even though the SBS concentration was raised. They attribute this phenomenon to the existence of particular sulfur-oxidizing bacteria (SOB) that can utilize bisulfite or sulfite as a sole energy source. It was also found that several bacteria in the brine were grown in a defined inorganic medium with the intermittent addition of SBS. The three isolated SOBs were Thiobacillus-like and facultatively autotrophic bacteria that are similar to the most general SOB in the sea. The same result was reported by Takeuchi et al. [356]. It was found that SOB was dominant in RO feed water after SBS dosing using denaturing gradient gel electrophoresis (DGGE) and sequencing technique. SOB uses SBS as a sole energy source in RO desalination plants. Thus excess SBS dosing caused a rapid growth of SOB.

The last issue about an SBS role triggering biofouling is the possibility of increasing the feed water’s assimilable organic carbon (AOC). As mentioned in Section 8, strong oxidants are generated from SBS and decompose organic compounds under certain conditions. An anomalous TOC increase was first observed in an ultrapure water (UPW) production [357]. The primary UPW system consists of a two-bed, three-tower pure water system (2B3T), RO, a mixed bed ion exchange (MB), and a vacuum degasifier. SBS was added to eliminate residual chlorine so that the residual sodium hypochlorite added to a media filter would not flow into the 2B3T.

When the operation was started, the 2B3T outlet conductivity was maintained for 24 h, as designed. On the other hand, TOC began to increase from about 8 h after starting water supply following to regenerating the ion exchange resins. The TOC concentration of the entire system rose as well. When SBS injection was stopped, the TOC did not increase, and the TOC concentration of the whole system was stabilized. More than 100 ppb of TOC difference was observed at the outlet of the anion tower between the SBS addition and a no addition case.

In the SWRO area, Weinrich et al. [358,359,360] observed the AOC increase after SBS dosing at the Tampa Bay seawater desalination plant (TBSDP). It has been accepted that AOC can be used as a good indicator of RO biofouling. In TBSDP, seawater was pretreated with approximately 0.5–1.1 mg/L as the Cl_2_ of chlorine dioxide at first, and then sulfuric acid, hypochlorite, and ferric chloride were injected. After a conventional coagulation–media filtration, diatomaceous earth (DE) filters and cartridge filters were placed prior to the SWRO desalination membranes. AOC was generally below detection (<10 µg/L) after DE filtration. However, after the cartridge filter and dosing SBS, AOC increased to 97 ± 19 µg/L in September and 23 ± 1 µg/L in October 2012. Records indicating higher SBS doses in October and November coincided with the periods in which differential pressure increased. Martorell et al. [79] reported the unusually high ORP values within the feed to the RO trains with no free chlorine concentration detected and overdosing SBS (20 ppm) in TBSDP.

Another anomalous DOC increase was observed in the large desalination plant located on the Red Sea coast in Saudi Arabia. Khan et al. [361] characterized the TOC/DOC in various pretreatment steps. In this plant, SWRO membranes were also exposed to chlorine (0.25–0.30 mg/L) by stopping SBS dosing (1.5–2.5 mg/L) in the feed stream for 1 h after every seven hours of operation. When conducting the liquid chromatography—organic carbon detection (LC-OCD) analysis, TOC/DOC contents increased after the SBS dosing sample. LC-OCD chromatogram confirmed an increase of the signal in the region assigned to medium to lower molecular weight organics. The increment in medium to lower molecular weight organics was also observed in the second sampling event. However, the authors suspected that the cartridge filter was in or near the saturation state and thus started leaching organics. Another possible route to increase AOC is reactions between SBS and DBPs. Yang et al. [362] suggested that the concentrations of THMs and haloacetic acids (HAAs) in the UF effluent slightly decreased by 9.0% and 3.7%, respectively, after the following addition of SBS, which might be attributed to the reaction between sulfite and DBPs in which some DBPs could be destroyed by reaction with sulfite.

To minimize the risk of biofouling triggered by SBS, controlling residual SBS concentration is of importance. There have been reports that implementing this practice could solve the biofouling problem. Hirai et al. [356,363] evaluated the effect of residual SBS on biofouling at an SWRO demonstration plant in Dukhan, Qatar. They expressed the residual SBS concentrations as the residual chlorine. The differential pressure demonstrated a decreasing trend when the SBS residue was decreased to a value equivalent to 0.1 mg/L chlorine (0.15 mg/L SBS). When the residual SBS was adjusted to the equivalent value of 0.2 mg/L (0.3 mg/L SBS) for safety, the change of differential pressure almost disappeared. Byrne [52] suggested maintaining a residual sulfite concentration greater than zero but less than 2 mg/L as SBS. As mentioned, the Santa Barbara seawater desalination plant in Curacao experienced heavy biofouling. After the SBS addition program was modified to reduce the excess, the rate of DP increase was immediately affected [62].

Up to this point, three possibilities demonstrating that SBS enhances biological growth and biofouling in the RO process were discussed. However, as Kurihara et al. [364] indicated, no data show direct evidence and quantitative impact of chlorination–dechlorination on SWRO biofouling. Thus, the quantitative influence and mechanisms of chemical additives on biofouling need to be clarified.

## 10. Conclusions

RO is now ubiquitous in water treatment and has been used for various applications. Innovative membrane development was a key to realizing commercial-scale RO plants. However, to be commercially successful in RO, accumulating a lot of additional science and technological developments was necessary. In that sense, RO is considered to be a highly integrated system consisting of a series of unit processes: (1) intake system, (2) pretreatment, (3) RO system, (4) post-treatment, and (5) effluent treatment. In each step, a variety of chemicals are used. In this review, the roles of sulfites as one of the essential chemicals are attempted to be summarized.

As for the established usages of SBS, such as dechlorination, shock treatment, deoxygenation, and preservation, the author strived to clarify the historical background and was happy to touch on some historical milestones, including the Coalinga BWRO plant, polyamide hollow fiber RO, PEC-1000 polyether RO membranes, dechlorination, and shock treatment.

Although sulfites are essential chemicals in RO, they have some adverse effects on RO membranes and processes. In particular, the RO membrane oxidation catalyzed by heavy metals and a trigger of biofouling are critical issues to achieve stable operations. This review shed light on the mechanism of membrane oxidation and triggering biofouling by sulfites. Generating strong oxidants, such as ∙SO_4_^−^ and ∙OH radicals, in the presence of oxygen and heavy metals was identified as one of the root causes of membrane oxidation and triggering biofouling. The generated oxidants attack RO membranes directly or decompose feed DOC, which results in increasing AOC concentration. One of the measures to prevent such problems is rigorously monitoring the residual SBS concentration and minimizing the SBS dosing. However, at this moment, directly measuring SBS concentration is rarely implemented in actual plants. Thus, it may be necessary to monitor the concentration with an SBS monitor in the future.

Odor is another concern from the point of occupational safety and health when using SBS. In that sense, odorless chemicals have been developed to replace SBS as a preservative. As the chemical-free desalination process has been investigated, the roles of SBS might be shortly changed.

Finally, if the author could fill one piece of the integrated RO technology jigsaw puzzle, this author would be unexpectedly happy.

## Figures and Tables

**Figure 1 membranes-12-00170-f001:**
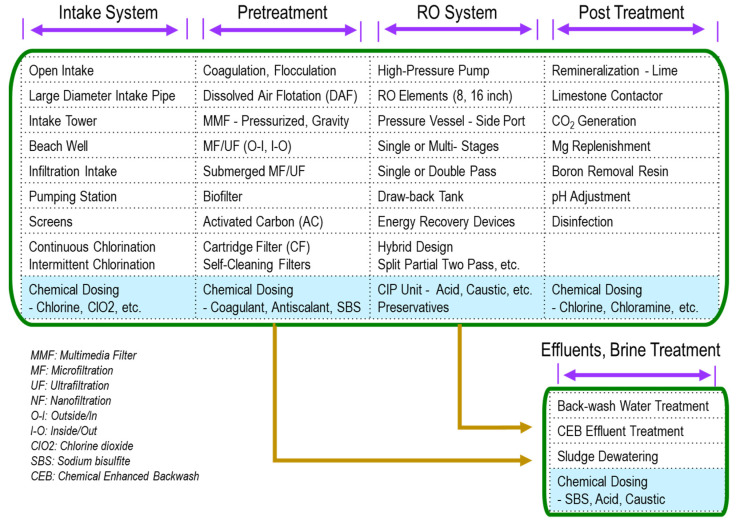
Key unit processes of a seawater desalination RO system and chemical usage.

**Figure 2 membranes-12-00170-f002:**
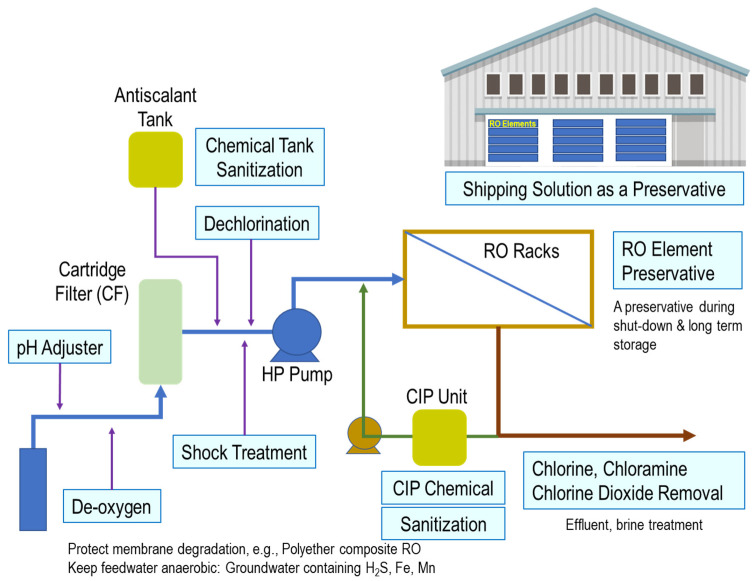
SMBS/SBS applications for RO systems.

**Figure 3 membranes-12-00170-f003:**
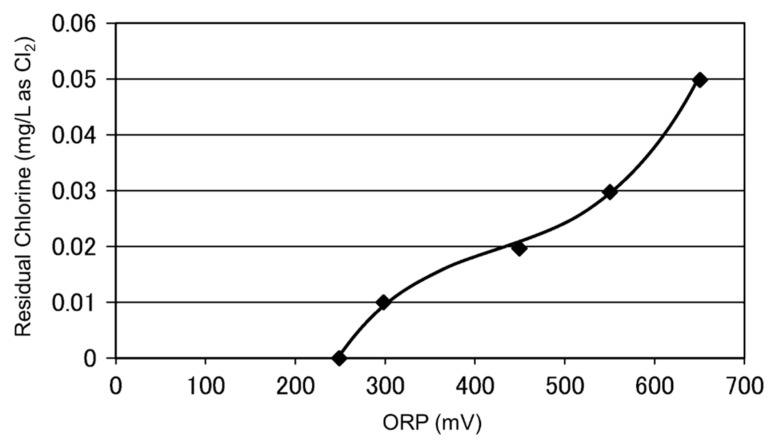
Relationship between ORP and free chlorine concentration in seawater [66].

**Figure 4 membranes-12-00170-f004:**
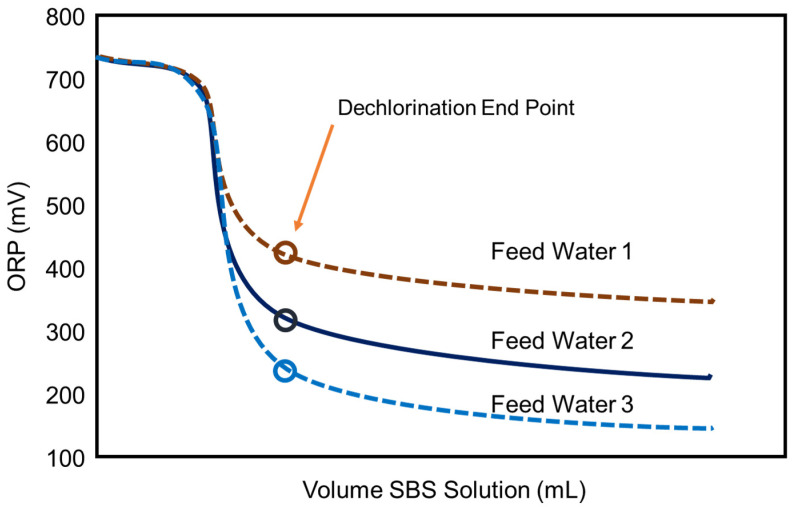
Schematic dechlorination titration curves by SBS injection.

**Figure 5 membranes-12-00170-f005:**
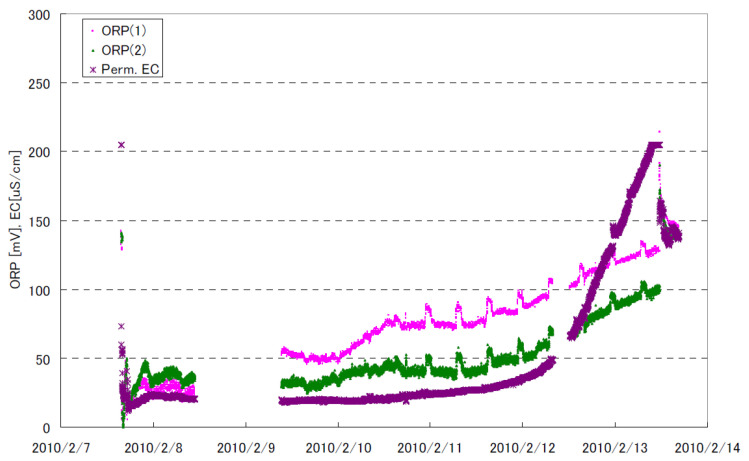
Operation data of the degradation in the Shuqaiq desalination plant [255].

**Figure 6 membranes-12-00170-f006:**
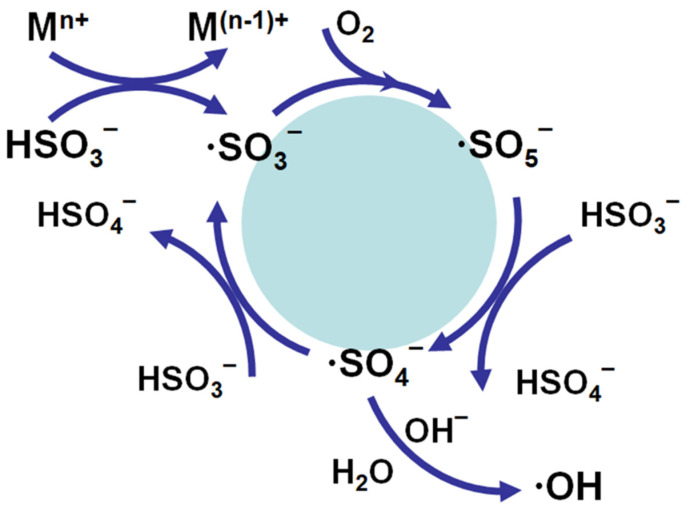
Major reactions of transition metal–bisulfite system [260,286,289,302].

**Figure 7 membranes-12-00170-f007:**
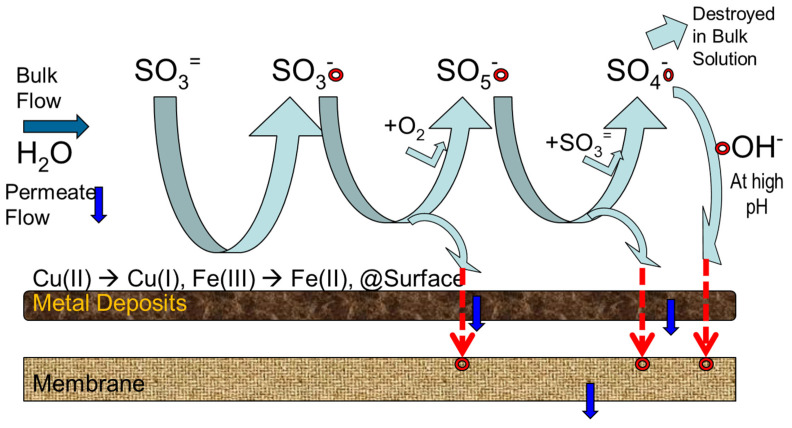
Proposed radical side-stream that arrives at RO membrane surface [255].

**Figure 8 membranes-12-00170-f008:**
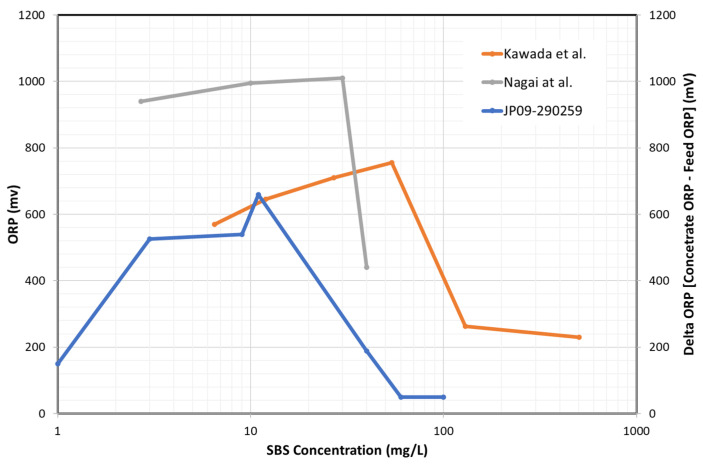
ORP as a function of dosing SBS concentration. Kawada et al. [309]: 3.5% NaCl + 146 ppm NaHCO_3_ + 400 ppb Cu^2+^. Nagai et al. [248]: Seawater + 100 ppb Cu^2+^, temperature 32 °C and pH 6.5. JP09-290259 [257]: Seawater + 1.7 ppb Cu^2+^ (RO concentrate).

**Figure 9 membranes-12-00170-f009:**
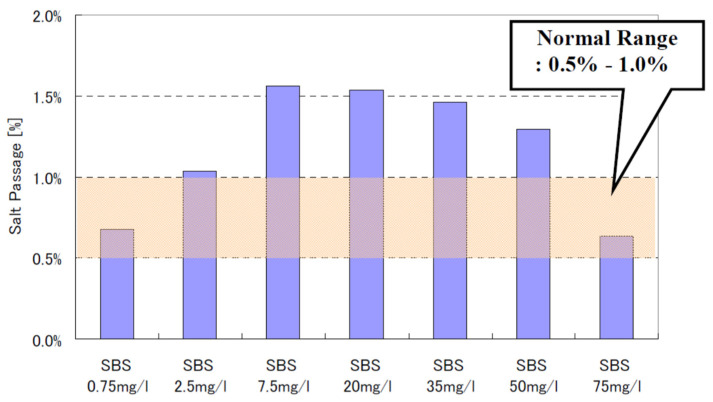
Effect of SBS concentration on salt passage of BWRO membrane [255].

**Figure 10 membranes-12-00170-f010:**
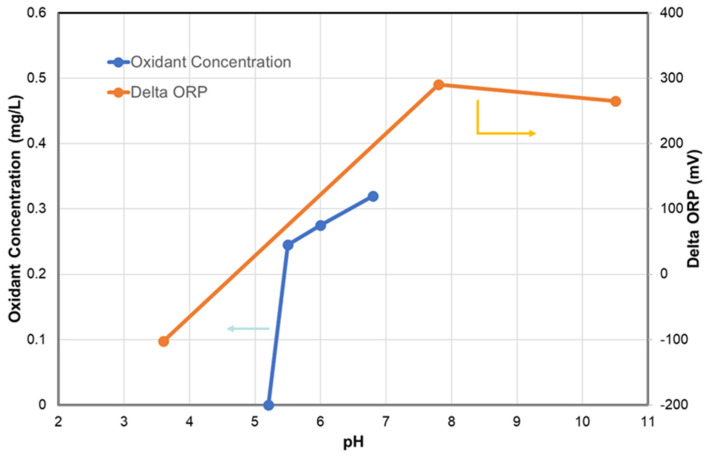
ORP changes and oxidant concentration as a function of pH [114,309]. Oxidant concentration [114]: RO brine water + 0.1 ppm ClO_2_ + chlorite 0.64 ppm + 10 ppm SBS. Delta ORP [309]: 3.5% NaCl + 146 ppm NaHCO_3_ + 400 ppb Cu^2+^ + 11 ppm SBS.

**Table 1 membranes-12-00170-t001:** SBS identification and physicochemical properties.

IUPAC Name	Sodium Hydrogen Sulfite
CAS Number	7631-90-5
Molecular Formula	NaHSO_3_
Molar Mass	104.06 g/mol
Solubility in Water	42 wt% in water 20 °C
Solution Density (20 °C)	1.304 (37 wt% aq. Solution) 1.360 (42 wt% aq. Solution)
Odor	A slight odor of sulfur dioxide

**Table 2 membranes-12-00170-t002:** Shelf life of SBS solutions of various concentrations [22,23,24].

Solution (wt%)	Shelf Life
2	Three (3) days
10	One (1) week
20	One (1) month
30	Six (6) months

**Table 3 membranes-12-00170-t003:** Theoretical sulfites dosages for dechlorination [21,37].

Sulfites	Molecular Weight	Theoretical Dosage to Remove 1 mg Chlorine (mg)
Sodium sulfite	126.1	1.78
Sodium bisulfite (SBS)	104.1	1.46
Sodium metabisulfite (SMBS)	190.2	1.34

**Table 4 membranes-12-00170-t004:** The suggested dosing amount of sulfites compounds for dechlorination.

Sulfites	Dosage to Remove 1 mg Chlorine (mg)	Stochiometric Amount Ratio	Comments	Reference
SMBS	3 >3	2.24 >2.24	Brackish water Seawater	[23]
	3	2.24		[41,42]
	2	1.49		[24]
SBS		3–5	Aramid polyamide	[43]
		>2		[44]
	2	1.37		[45]

**Table 5 membranes-12-00170-t005:** The reported threshold ORP values to avoid membrane oxidation.

Membrane Manufacturer	High Alarm H (mV)	High-High Alarm HH (mV)	Reference
A	-	175–200	[68]
B	300	350	[69]
C	250	300	[70]
D		270 (at pH = 6.0)	[71]
		200 (at pH = 8.0)	

**Table 6 membranes-12-00170-t006:** Disinfectants removal needs in RO process.

Disinfectant	Redox Potential (V)	Feed	Permeate	Concentrate	Discharge from CIP Sanitization
Chloramine	0.75	✓	—	✓	—
Chlorine dioxide	0.95	✓	In case	✓	—
DBNPA	─	—	NA	✓	✓

Note: In the case of ClO_2_, chlorite and chlorate may have to be removed.

**Table 7 membranes-12-00170-t007:** Three scenarios of bromamine formation in seawater.

Scenario	Related Key Reactions	Formed Chloramines Bromamines
Prechloirnated SW + NH_4_ salts injection	Br^Ȓ^ + HOCl → Cl^−^ + HOBr NH_3_ + HOBr → NH_2_Br + H_2_O NH_2_Br + HOBr → NHBr_2_ + H_2_O	Monobromamine Dibromamine
NH_4_ salts first or NH_4_ salts and NaOCl injection to SW together	NH_3_ + HOCl → NH_2_Cl + H_2_O	Monochloramine
	NH_2_Cl + HOCl → NHCl_2_ + H_2_O	Dichloramine
	NH_3_ + HOBr → NH_2_Br + H_2_O	Bromamine
	NH_2_Br + HOBr → NHBr_2_ + H_2_O	Dibromamine
Preformed chloramine injection to SW	NH_2_Cl (stable for an hour in SW)	Monochloramine
	NH_2_Cl + Br^−^ + H^+^ → NHBrCl + NH_4_ + Cl^−^	Bromochloramine

**Table 8 membranes-12-00170-t008:** Theoretical sulfites and thiosulfate dosages for chlorine dioxide removal.

Sulfites	Molecular Weight	Theoretical Dosage to Remove 1 mg ClO_2_ (mg)
Sodium sulfite	126.1	4.67
SBS	104.1	3.85
SMBS	190.2	3.52
Sodium thiosulfate	158.11	1.46

**Table 9 membranes-12-00170-t009:** Oxidation potential for relevant oxidants.

Chemical Species	Standard Oxidation Potential (V)	Relative Strength
Hydroxy radical (∙OH)	2.8	2.0
Sulfate radical (∙SO_4_^−^)	2.6	1.8
Ozone (O_3_)	2.1	1.5
Persulfate Anion (S_2_O_8_^2−^)	2.0	1.4
Peroxymonosulfate (HSO_5_^−^) [307]	1.8	1.3
Chlorine (Cl_2_)	1.4	1.0
Peroxymonosulfate radical (∙SO_5_^−^) at pH 7.0 [308]	1.1	0.8
Sulfite radical (∙SO_3_^−^) at pH > 7.0 [308]	0.63	0.5

**Table 10 membranes-12-00170-t010:** Factors affecting membrane degradation by SBS.

Factors Affecting Membrane Degradation by SBS	References
Heavy metals	[47,248,252,309]
SBS concentration	[248,309]
Dissolved oxygen concentration	[248]
Feed pH	[47,197,309]
Bicarbonate ion concentration	[47,197,309]
Chloride ion concentration	[248]
Salinity or feed TDS	[47,197]
Temperature	[47,197]

**Table 13 membranes-12-00170-t013:** Effect of chelating agents for SWRO membrane performance [184].

Chelating Agent	Initial		100 h		1000 h	
	SP (%)	Flux (m/d)	SP (%)	Flux (m/d)	SP (%)	Flux (m/d)
None	0.52	0.68	0.78	0.69	1.60	0.81
SHMP 10 ppm	0.50	0.67	0.49	0.66	0.53	0.65
EDTA 5 ppm	0.49	0.69	0.51	0.68	0.54	0.65

Feed: 3.5% NaCl solution containing 10 ppb of Cu and 20 ppm of SBS. Test conditions (pressure): 56 kg/cm^2^. Flux (m/d): m^3^/m^2^/day.

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
