# Peer review of "Roles of Sulfites in Reverse Osmosis (RO) Plants and Adverse Effects in RO Operation"

_membranes, 2022, doi:10.3390/membranes12020170_

Round 1

Reviewer 1 Report

The review manuscript titled “Roles of Sulfites in Reverse Osmosis (RO) Plants and Adverse Effects in RO Operation” and written by Yasushi Maeda is well written and structured and report a comprehensive review on the effect of sulfites dosing in RO desalination plants. I recommend a minor revision based on the following comments:

  1. Page 2, lines 59-65. The author mentioned the chemicals used in RO systems. The author should remark that the use of these chemicals depends strongly on the feedwater characteristics and operating conditions. For example, there are some studies that have reported long-term operation of BWRO desalination plants only dosing antiscalant and carrying out chemical cleaning in place (80,000 h operational experience and performance analysis of a brackish water reverse osmosis desalination plant. Assessment of membrane replacement cost, Evaluation of the first nine years operating data of a RO brackish water desalination plant in Las Palmas, Canary Islands, Spain).

  1. Please, revise the list of references, there are some duplicities. For example, refs 26 and 331 are the same.

  1. Page 3, line 83, The abbreviation of sodium bisulfite (SBS) was already done in page 2, so please, use only the abbreviation in the rest de the document. Please, revise the same issue for the rest of the abbreviations.

  1. In the text, when reference to a table is written it should be written Table instead of table. Please, revise the entire manuscript. (Page 6, line 200)

  1. Page 8, line 290. Please, write m3/d instead of m3/day. Revise the entire document

  1. In Figure 3, Y-axe label. Please, write 2 as subscript.

  1. If Figure 7, was not self-elaborated please, add the reference in the caption of the figure.

  1. Page 37, line 1546, please write pH=10 as it is written in the rest of the document. Revise the entire document

Author Response

Response to Reviewer 1 Comments

Point1. Page 2, lines 59-65. The author mentioned the chemicals used in RO systems. The author should remark that the use of these chemicals depends strongly on the feedwater characteristics and operating conditions. For example, there are some studies that have reported long-term operation of BWRO desalination plants only dosing antiscalant and carrying out chemical cleaning in place (80,000 h operational experience and performance analysis of a brackish water reverse osmosis desalination plant. Assessment of membrane replacement cost, Evaluation of the first nine years operating data of a RO brackish water desalination plant in Las Palmas, Canary Islands, Spain).

 Response 1: The following sentence was added:

However, it should be noted that the use of these chemicals depends strongly on the feedwater characteristics and operating conditions. For example, some well water treatment plants are only equipped with cartridge filters with minimal chemical dosage. It is reported that the 360 m3/d capacity BWRO plant in Las Palmas, Canary Islands, Spain, has been operating for more than nine years by only dosing 6 mg/L of antiscalant [Ruiz-García et al. 2015]. Similarly, Lagartos et al. [2019] reported Malta's Pembroke seawater desalination plant. The plant can produce 54,000 m3/d of water. The water intake comes from beach wells with a silt density index below 1. The pH is adjusted with sulfuric acid down to 6.7 to protect the pipework and prevent scaling. After pH adjustment, cartridge filters are installed upstream of the RO unit. It is reported that the cleaning in place (CIP) frequency varies between 6 and 10 months, depending on the train condition and time of operation.

Point2. Please, revise the list of references, there are some duplicities. For example, refs 26 and 331 are the same.

 Response 2: Revised this article as shown below:

Idais, R. H.; Abuhabib, A. A.; Hamzah, S. (2021). Recent Advances in Measuring and Controlling Biofouling of Seawater Reverse Osmosis SWRO: A Review. 2021, Osmotically Driven Membrane Processes.

Point3. Page 3, line 83, The abbreviation of sodium bisulfite (SBS) was already done in page 2, so please, use only the abbreviation in the rest de the document. Please, revise the same issue for the rest of the abbreviations.

 Response 3: Chnaged all “sodium bisulfite” to SBS and “sodium metabisulfite” to SMBS.

Point4. In the text, when reference to a table is written it should be written Table instead of table. Please, revise the entire manuscript. (Page 6, line 200)

 Response 4: Revised this as “Table” and checked the entire manuscript.

Point5. Page 8, line 290. Please, write m3/d instead of m3/day. Revise the entire document

 Response 5: Changed to m3/d in the entire document, except reference

Point6. In Figure 3, Y-axe label. Please, write 2 as subscript.

 Response 6: Rewrite 2 as subscript

Point7. If Figure 7, was not self-elaborated please, add the reference in the caption of the figure.

 Response7: Indicated original references.

Point8. Page 37, line 1546, please write pH=10 as it is written in the rest of the document. Revise the entire document

Response 8: Revised to pH=10

Reviewer 2 Report

The authors in this review manuscript mainly considered the influence of sulfites on the RO plants and operations. They summerized the key processes of RO treatment especially for chemical processes, and corresponding materials used within RO membranes with considering the unfoiling properties. The manuscript is of great significance in directing design of RO systems. Nevertheless, I have two suggestions in both References and Conclusion sections.

Firstly, some important novel mechanism and related publications in RO and desalination fields should be cited, e.g., Science Advances (2020) 6: eaba9471;  Science  (2019) 364: 1033–1034.

Secondly, the authors can clarify the perspectives in theory, mechanism, and applications in RO domain associated with the current challenges in Conclusion section.

Author Response

Firstly, some important novel mechanism and related publications inRO and desalination fields should be cited, e.g., Science Advances(2020) 6: eaba9471; Science (2019) 364: 1033–1034.

Based on you suggestion, I added brief description about RO membrane progress as follows:

Regarding RO membrane development, more than 60 years have passed since UCLA first announced the development of an innovative asymmetric cellulose acetate RO membrane in 1960. Furthermore, new generation polyamide hollow fiber RO and thin-film composite (TFC) aromatic polyamide RO membranes were developed one after another in the early 70s and 1977. As a result of continuous improvements, the TFC RO membrane performance has been greatly improved, and it is now widely used for a variety of applications. As for the future membrane desalination technology, three technologies were raised in National Geographic, April 2010 [xx]. These three technologies promised to reduce the energy requirement of desalination up to 30% are forward osmosis, carbon nanotubes, and biomimetics. Among those, nanoporous membranes, including porous graphene, carbon nanotubes, graphene oxide, etc., attracted much attention from academic researchers [Zhang et al. 2020]. However, it doesn't seem easy to produce commercial-based defect-free RO membranes with nanoporous materials. A way of overcoming material limitations for RO applications is to utilize composite materials comprising nanoporous materials within a polymer matrix. The use of thin-film nanocomposite (TFN) membranes for water purification was first described for BWRO membranes by Hoek et al. [Hoek et al., 2007]. After that, many research works on the TFN membranes have been conducted [Bassyouni 2019, Wen 2019].

Secondly, the authors can clarify the perspectives in theory, mechanism, and applications in RO domain associated with the current challenges in Conclusion section.

Acoording to your guidance, I added concise mechanism of membrane oxidation as follows:

Although sulfites are essential chemicals in RO, they have some adverse effects on RO membranes and processes. Especially, the RO membrane oxidation catalyzed by heavy metals and a trigger of biofouling are critical issues to achieve stable operations. This review shed light on the mechanism of membrane oxidation and triggering biofouling by sulfites. Generating strong oxidants such as žSO4 and žOH radicals in the presence of oxygen and heavy metals was identified as one of the root causes of membrane oxidation and triggering biofouling. The generated oxidants attack RO membranes directly or decompose feed DOC, resulting in increasing AOC concentration.